# D-MiSo: Editing Dynamic 3D Scenes using Multi-Gaussians Soup

**Joanna Waczyńska**[*†]
Doctoral School of Exact and Natural Sciences
Jagiellonian University

**Piotr Borycki**[*]
Faculty of Mathematics and Computer Science
Jagiellonian University

**Joanna Kaleta**
Warsaw University of Technology
Sano Centre for Computational Medicine

**Sławomir Tadeja**
Department of Engineering
University of Cambridge

**Przemysław Spurek**
Jagiellonian University
IDEAS NCBR

## Abstract

Over the past years, we have observed an abundance of approaches for modeling dynamic 3D scenes using Gaussian Splatting (GS). These solutions use GS to represent the scene's structure and the neural network to model dynamics. Such approaches allow fast rendering and extracting each element of such a dynamic scene. However, modifying such objects over time is challenging. SC-GS (Sparse Controlled Gaussian Splatting) enhanced with Deformed Control Points partially solves this issue. However, this approach necessitates selecting elements that need to be kept fixed, as well as centroids that should be adjusted throughout editing. Moreover, this task poses additional difficulties regarding the re-productivity of such editing. To address this, we propose **D**ynamic **M**ult**i**-Gaussian **So**up (D-MiSo), which allows us to model the mesh-inspired representation of dynamic GS. Additionally, we propose a strategy of linking parameterized Gaussian splats, forming a Triangle Soup with the estimated mesh. Consequently, we can separately construct new trajectories for the 3D objects composing the scene. Thus, we can make the scene's dynamic editable over time or while maintaining partial dynamics.

## 1 Introduction

Recently introduced Gaussian Splatting (GS) [1] represents the 3D scene structure through Gaussian components. We can combine GS with the neural network (i.e., deform network) to model dynamic scenes [2]. This approach involves the joint training of both the GS components and the neural network. GS characterizes the 3D object's shape and color, while the neural network utilizes time embedding and Gaussian parameters to generate updated initial positions to model dynamic scenes. Such an approach allows for fast rendering and extracting each element of a dynamic scene.

Most existing methods can effectively model dynamic scenes, but generating new 3D objects' positions remain challenging. Consequently, we cannot edit objects over time when using such

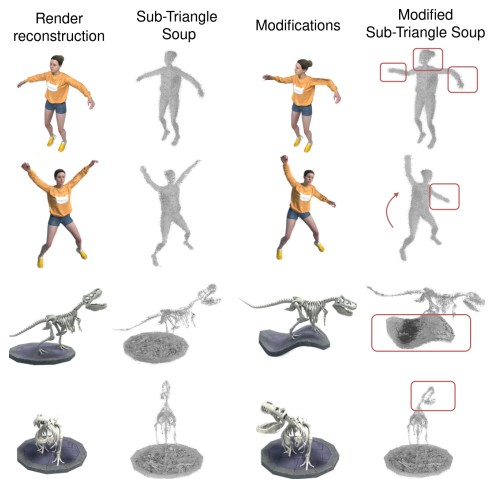

Figure 1: D-MiSo model parameterized dynamic scenes by Triangle Soup (disjoint triangles cloud), which allows modification of objects during time.

---

[*]Equal contribution
[†]`joanna.waczynska@doctoral.uj.edu.pl`

38th Conference on Neural Information Processing Systems (NeurIPS 2024).

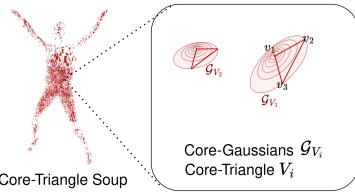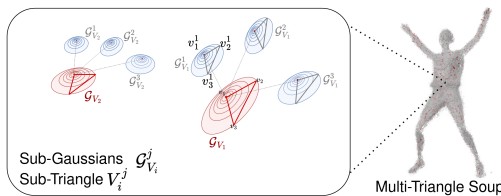

Figure 2: Each object using the D-MiSo model is represented by Core-Gaussians and Sub-Gaussians, which form Multi-Gaussians. Each Gaussian is related to a triangle using parameterization proposed in GaMeS [3]. Triangles define the Gaussian shape (i.e., location, scale, rotation), and triangles clouds form Triangles Soups.

approaches. To tackle this issue, SC-GS (Sparse Controlled Gaussian Splatting) [4] uses Deformed Control Points to manage Gaussians. After the training phase, we can manually modify the model at any point in time. However, this method requires identifying elements to remain static and adjusting 3D objects' centroids (nodes) during editing when a relationship between the selected nodes is visible. For example, by moving the humanoid 3D model's hand, the part of the head or leg is also changed.

To address this issue, we introduce **D**ynamic **M**ulti-Gaussian **So**up (D-MiSo), which is easier to modify (Fig. 1) and obtain renders comparable to SC-GS. D-MiSo estimates the mesh as Triangle Soup, i.e. a set of disconnected triangle faces [5, 6], and uses a dynamic function to control the vertices.

D-MiSo employing Multi-Gaussians, defined as larger Core-Gaussians encompassing smaller ones termed Sub-Gaussians (Fig. 2). Sub-Gaussians are defined in the local coordinate system given by principal components of Core-Gaussian. Therefore, by modifying Core-

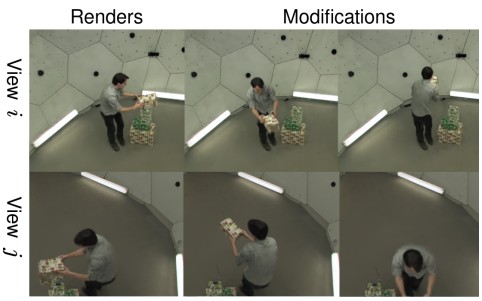

Figure 3: D-MiSo allows us to modify scenes in similar ways as classical mesh-based models.

Gaussian, we change all Sub-Gaussians, which allows scene modifications (Fig 3). Core-Gaussians are an alternative to the control points discussed in [4], with the added advantage of allowing individual modifications. Consequently, there is no necessity for static and dynamic markers. In Fig. 4, we present the difference between modification applied by SC-GS and D-MiSo.

Our model uses flat Gaussians. Therefore, based on GaMeS [3], we can approximate Gaussian components using triangle face mesh by parameterizing Gaussian components by the vertices of the mesh face. We denote such transformation by $\mathcal{T}(\cdot)$. In practice, as a consequence of parameterizing each Gaussian, we obtain a cloud of triangles called Triangle Soup [5]. Using Triangle Soup, we can control two types of Gaussian components. Accordingly, in D-MiSo, we get: Sub-Triangle Soup, Core-Triangle Soup and Multi-Triangle Soup (Fig. 2, Core-Triangle Soup is marked by red color, and Sub-Triangle Soup is denoted in blue). In D-MiSo, we can select and modify one part of the object like a mesh. In contrast, using SC-GS, static and dynamic points have to be selected, and editing only one part of the object is difficult.

During training, the positions of Core-Gaussians are managed by deformation multilayer perceptron (MLP), while the Sub-Gaussians are collectively manipulated through global transformation and small local deformation. The former describes the general flow of objects in the scene, while the local deformation is responsible for modeling small changes like shadows and light reflections. After training, we can modify our model directly by using the vertex of the Sub-Triangle Soup, or we can generate mesh from the Core-Triangle Soup (Fig. 5).

The contributions of this paper are significant and are outlined as follows:

- We introduce the Multi-Gaussian components, which consist of a single large Gaussian response for global transformations and many small components dedicated to rendering. Multi-Gaussian components allow for the modeling of large 3D scene elements.

- We propose D-MiSo a model that uses Multi-Gaussian components and two deformation networks for modeling dynamic scenes.

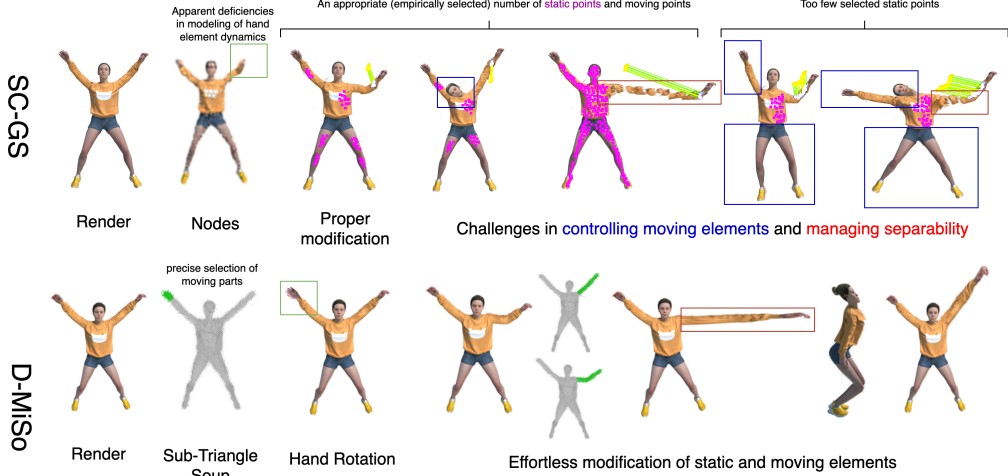

Figure 4: Comparison of possible modifications in D-MiSo and the SC-GS. In the latter, authors use nodes while D-MiSo apply Sub-Triangle Soup (see the second column). We also must add static (pink) and dynamic (yellow) points in SC-GS to obtain modification by editing dynamic points. In practice, we have to use many static points to stop artifacts. Moreover, SC-GS is not an affine invariant and produces space when we change the size of the objects. In the case of D-MiSo, we marked points and applied modifications. Our model is superior in handling object scaling.

- Our D-MiSo allows an object to be edited at a selected moment in time. The edited components are independent, and the editing does not affect other parts of the object. In addition, it also allows for full or partial dynamics to be maintained. Modifications also include scaling and rotation.

## 2 Related Works

Recent advancements in view synthesis, particularly driven by NeRFs [7], have significantly contributed to the rapid development of novel view synthesis techniques. However, the majority of these approaches model static scenes implicitly using MLP. Moreover, several works have extended classical NeRF to dynamic scenes through the use of deformation fields [8, 9, 10] and [11]. The alternative approaches, such as [12] and [13], represent scenes as 4D radiance fields. Early works on dynamic scenes face difficulties when dealing with monocular settings and uncontrolled or lengthy scenarios. To enhance scene motion modeling, some works utilize flow-based techniques [10, 14, 15]. However, NeRF-based solutions often suffer from long training and rendering times. To address this, grid-plane-based methods [16, 17, 18] have been proposed. In addition, several NeRF-based approaches have also been extended for scene editing purposes [19, 20, 21, 22].

The recently introduced Gaussian Splatting (GS) technique [1] addresses many limitations of other methods, offering multiple advantages due to their explicit geometry representation, enabling easier dynamics modeling. The efficient rendering of the 3D version of GS also avoids densely sampling and querying neural fields, making downstream applications such as free-viewpoint video reconstruction more feasible. A notable extension of 3D GS was proposed in [23], where the authors introduced the concept of anchor points to tackle the problem of overfitting caused by redundant Gaussians. Scaffold-GS addresses this issue by distributing local 3D Gaussians according to anchor points. This approach reduces redundancy, enhances scene coverage, and maintains high-quality rendering with improved robustness to view changes. While the original GS was developed for static scenes, several extensions for dynamic scenes were proposed. Most of the early works operate in multiview setup [24, 25, 26]. For example, [26] utilizes a frame-by-frame approach to model each timestep. However, this method lacks inter-frame correlation and requires high storage overhead for long-term sequences. In [27, 2], MLP is introduced to model changes in Gaussians over time, and in [28] MLP together with decomposed neural voxel encoding algorithm are utilized for training and storage efficiency. In [29], dynamic scenes are divided into dynamic and static parts, optimized separately and rendered together to achieve decoupling. [24] spacetime Gaussian proposes approximating the spatiotemporal 4D volume of a dynamic scene by optimizing a collection of 4D primitives with explicit geometry and appearance modeling. This method uses 4D Gaussians parameterized by anisotropic ellipses and

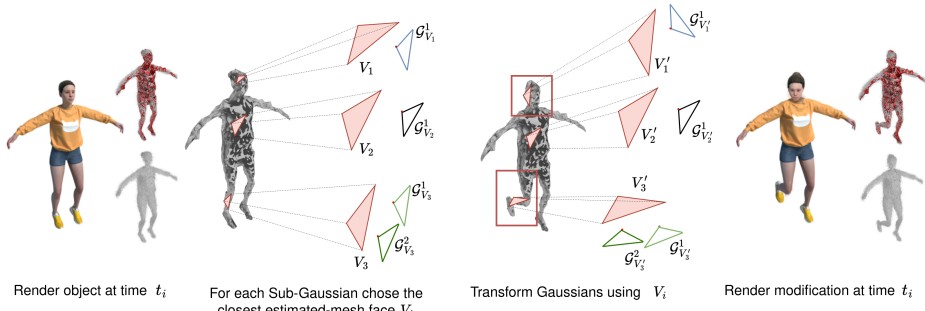

| Render object at time $t_i$ | For each Sub-Gaussian chose the closest estimated-mesh face $V_i$ | Transform Gaussians using $V_i$ | Render modification at time $t_i$ |

Figure 5: One way to modify the object at the selected time $t_i$ is to take Core-Gaussians and apply a meshing strategy to obtain the correct mesh instead of Triangle Soup. Then, we can parametrize Sub-Gaussian in the coordinate system given bay mesh faces instead of Core-Triangle Soup. Finally, we can modify our mesh to obtain new modifications.

view-dependent, time-evolved appearances represented by 4D spherical harmonics coefficients. In [25], a novel, real-time and photorealistic scene representation called Spacetime Gaussian Feature Splatting has been introduced. This approach extends 3D Gaussians with temporal opacity and parametric motion/rotation, enabling the capture of static, dynamic, and transient scene content. Additionally, the method incorporates splatted feature rendering to model view- and time-dependent appearances while maintaining a compact representation size. The method is notable for its high rendering quality and speed while also being storage-efficient. Other works enhance dynamic scene reconstruction using external priors. For example, the diffusion priors can be used as regularization terms during optimization [30].

Furthermore, GS was employed for mesh-based scene geometry editing. In [31], 3D Gaussians are defined over an explicit mesh and utilize mesh rendering to guide adaptive refinement. This approach depends on the extracted mesh as a proxy and fails if the mesh cannot be extracted. In contrast, in [32], explicit meshes are extracted from 3D GS representations by regularizing Gaussians over surfaces. However, this method involves a costly optimization and refinement pipeline. Another example of [33] employs sparse control points for 3D scene dynamics, but this method struggles with intense edit movements and necessitates accurate static node selection. Also, [3] combines GS with mesh extraction. However, such an approach only works for static scenes.

The method proposed in [34] combines meshes with GS and reconstructs a high-fidelity and time-consistent mesh from a single monocular video. However, it relies on a Poisson Solver and differentiable Marching Cubes to recover the deformed surface, significantly complicating the pipeline. Moreover, it does not explore geometry modification capabilities, which constitute a significant aspect of our work. Conversely, cage-based methods [35, 36] are an intuitive tool for geometry manipulation. However, they require additional steps for cage-building and may lack the flexibility and precision of manual, vertex-level deformation techniques, potentially missing fine details.

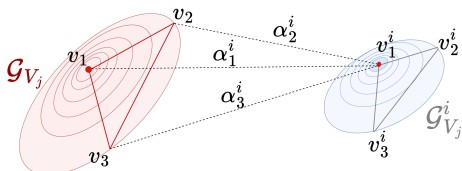

Figure 6: Multi-Gaussian, consisting of one Core-Gaussian $\mathcal{G}_{V_j}$ and a Sub-Gaussian $\mathcal{G}^i_{V_j}$. The Core Gaussian is parametrized by a $V_j$-triangle, and the Sub-Gaussian by a $V^i_j$-triangle. The relative distance of the center of the Sub-Gaussian from the Core-Gaussian is indicated by $\boldsymbol{\alpha^i} = (\alpha^i_1, \alpha^i_2, \alpha^i_3)$.

In contrast to the listed approaches, we propose a D-MiSo, a mesh-based method specifically designed to handle dynamic scenes. D-MiSo leverages a straightforward pipeline of GS techniques to enable real-time editing of dynamic scenes.

## 3 Dynamic Multi-Gaussian Soup

Here, we present the main components of D-MiSo. We start with the classical GS to provide the foundations for our model. Next, we introduce the concept of Multi-Gaussians and describe how to

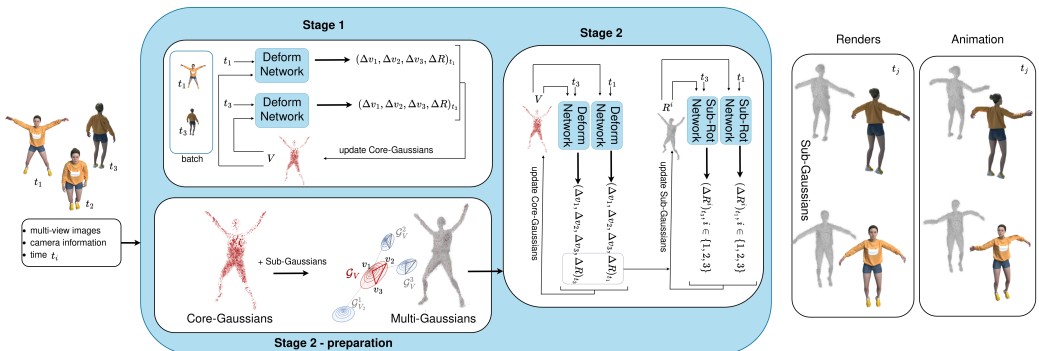

Figure 7: D-MiSo model diagram. The input of the model consists of images at different moments in time and information regarding the position of the camera. Training distinguishes two main phases (i.e., stages). The first includes the preparation of Core-Gaussians, describing the movement of the object. The second focuses on the fitting of Sub-Gaussians responsible for the render's quality. The model has the ability to produce high-quality renders or create an animation/modification of the object due to Sub-Gaussians (i.e., Sub-Triangles Soup) shape modification.

estimate the mesh for editing. Finally, we show D-MiSo, which uses Multi-Gaussians in dynamic 3D scenes.

**Gaussian Splatting**    The Gaussian Splatting (GS) technique models 3D scenes using an array of 3D Gaussians, each specified by its mean position, covariance matrix, opacity, and color expressed using spherical harmonics (SH) [37, 38]. The GS algorithm constructs the radiance field by iteratively optimizing the parameters of all Gaussian components. Ultimately, the GS efficiency mainly depends on its rendering method, which involves projecting Gaussian components.

The GS framework employs a dense collection of 3D Gaussians: $\mathcal{G} = \{(\mathcal{N}(\mathbf{m}_i, \Sigma_i), \sigma_i, c_i)\}_{i=1}^{n}$, where $\mathbf{m}_i$ denotes the position, $\Sigma_i$ the covariance, $\sigma_i$ the opacity, and $c_i$ the SH colors for the $i$-th Gaussian. The GS optimization process involves a repetitive cycle of rendering and comparing the resultant images with the training views. In our work, we will use Multi-Gaussian approaches (Fig. 6).

**Multi-Gaussians**    Multi-Gaussians $\mathcal{G}_{multi}$ are dedicated to describing relatively large parts of the 3D scene to allow modification of large blocks instead of modifying each small Gaussian separately. The Multi-Gaussian model comprises a primary large 3D Gaussian (referred to as the Core-Gaussian $\mathcal{G}_{core}$), which encompasses numerous smaller Gaussians (termed Sub-Gaussians $\mathcal{G}_{sub}$), all of which are parameterized by the main Core-Gaussian. Multi-Gaussianare is similar to anchor Gaussians from [39], but we do not use a neural network to produce child components. We parametrize Sub-Gaussians in a local coordinate system.

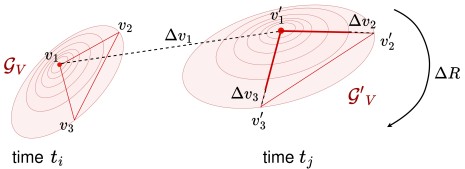

Figure 8: Representation of change over time acting on Core-Gaussians using a neural network responsible for movement. In practice, $t_i$ is an abstract time. The network's output returns information about the change in location $\Delta v_1$, scale ($\Delta v_2$, $\Delta v_3$), and rotation $\Delta R$.

Similarly to classical GS, we parameterize the Core-Gaussian distribution by center $\mathbf{m}$ and the covariance parameterized by factorization: $\Sigma = RSSR^T$, where $R$ is the rotation matrix and $S$ the scaling parameters. More precisely we consider $p$ Core-Gaussians uses flat Gaussians as in [3], and is defined by:

$$\mathcal{G}_{core} = \{(\mathcal{N}_{core}(\mathbf{m}_i, R_i, S_i), \sigma_i, c_i)\}_{i=1}^{p}, \tag{1}$$

where $S = \text{diag}(s_1, s_2, s_3)$, $s_1 = \varepsilon$ and $R$ is rotation matrix of Core-Gaussian defined as: $R = (\mathbf{r}_1, \mathbf{r}_2, \mathbf{r}_3)$, where $\mathbf{r}_i \in \mathbb{R}^3$ which can be interpreted as a local coordinate system used by Sub-Gaussian.

It is worth noting that Sub-Gaussian can be interpreted as a child of Core-Gausian. We define centers of Sub-Gaussian $\mathcal{N}_{sub}(\mathbf{m}^i, R^i, S^i)$ in the local coordinate system of Core-Gaussian $\mathcal{N}_{core}(\mathbf{m}, R, S)$ by: $\mathbf{m}^i = \mathbf{m} + R\boldsymbol{\alpha}^{iT}$, where $\mathbf{m}, R$ is Core-Gaussian position and rotation; and $\boldsymbol{\alpha}^i = (\alpha_1^i, \alpha_2^i, \alpha_3^i)$ are trainable parameters used to define the positions of the Sub-Gaussian relative to the Core-Gaussian (Fig. 6). Sub-Gaussians are used for rendering and can be seen as a main component of our model:

$$\mathcal{G}_{sub} = \left\{ \left( \mathcal{N}_{sub}\left(\mathbf{m} + R\boldsymbol{\alpha}^{iT}, R^i, S^i\right), \sigma^i, c^i \right) \right\}_{i=1}^{k}, \tag{2}$$

where $\mathbf{m}, R, S$ are parameters of Core-Gaussian and, $\boldsymbol{\alpha}^i, S^i, R^i$ marks parameters of $i$-th Sub-Gaussians with opacity $\sigma^i$, and SH colors $c^i$.

Core-Gaussians is generally dedicated mainly to transformations, hence, in practice opacity or colors are not used during rendering. On the other hand, Sub-Gaussian is devoted to rendering and modification. It has its own opacity and colors.

**GaMeS parametrisation of Multi-Gaussian component**    Multi-Gaussians describe 3D scenes using solid blocks rather than tiny Gaussian distributions. This method enhances our model's suitability for dynamic environments. One of our mode's most important properties is its ability to model dynamic scenes in each time step. To obtain such properties, we parameterize all Gaussian using Triangle Soup following similar approach as in GaMeS [3]. Thanks to a few simple transformations, we can convert the mean $\mathbf{m}$ rotation $R$ and scaling $S$ into triangle $V = (\mathbf{v}_1, \mathbf{v}_2, \mathbf{v}_3)$, which parametrizes Gaussian distribution. Such transformation is unambiguous and reversible.

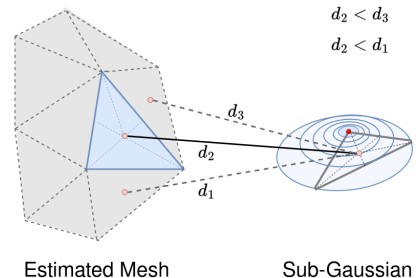

Estimated Mesh          Sub-Gaussian

Figure 9: One way to animate is to assign the nearest triangle (face) to each Sub-Gaussian from the estimated mesh. The mesh modification changes the assigned Gaussian.

Let us assume that we have a Gaussian component parameterized by mean $\mathbf{m}$, rotation matrix $R = [\mathbf{r}_1, \mathbf{r}_2, \mathbf{r}_3]$ and scaling $S = \text{dig}(\varepsilon, s_2, s_3)$. We define three vertex of a triangle (face): $V = [\mathbf{v}_1, \mathbf{v}_2, \mathbf{v}_3]$,, where $\mathbf{v}_1 = \mathbf{m}$, $\mathbf{v}_2 = \mathbf{m} + s_2\mathbf{r}_2$, $\mathbf{v}_3 = \mathbf{m} + s_3\mathbf{r}_3$.

Now we froze the vertex of the face $V = [\mathbf{v}_1, \mathbf{v}_2, \mathbf{v}_3]$ and reparameterize the Gaussian component by defining $\hat{\mathbf{m}}$, $\hat{R} = [\hat{\mathbf{r}}_1, \hat{\mathbf{r}}_2, \hat{\mathbf{r}}_3]$ and $\hat{S} = \text{dig}(\hat{s}_1, \hat{s}_2, \hat{s}_3)$. First, we put $\hat{\mathbf{m}} = \mathbf{v}_1$. The first vertex of $\hat{R}$ is given by a normal vector:

$$\hat{\mathbf{r}}_1 = \frac{(\mathbf{v}_2 - \mathbf{v}_1) \times (\mathbf{v}_3 - \mathbf{v}_1)}{\|(\mathbf{v}_2 - \mathbf{v}_1) \times (\mathbf{v}_3 - \mathbf{v}_1)\|},$$

where $\times$ is the cross product. The second one is defined by $\hat{\mathbf{r}}_2 = \frac{(\mathbf{v}_2 - \mathbf{v}_1)}{\|(\mathbf{v}_2 - \mathbf{v}_1)\|}$. The third one is obtained as a single step in the Gram–Schmidt process [40]:

$$\hat{\mathbf{r}}_3 = \text{orth}(\mathbf{v}_3 - \mathbf{v}_1; \mathbf{r}_1, \mathbf{r}_2).$$

Scaling parameters can also be easily calculated as $s_1 = \varepsilon$, $\hat{s}_2 = \|\mathbf{v}_2 - \mathbf{v}_1\|$ and $\hat{s}_3 = \langle \mathbf{v}_3 - \mathbf{v}_1, \hat{\mathbf{r}}_3 \rangle$. Consequently, the covariance of Gaussian distribution positioned on face is given by:

$$\hat{\Sigma}_V = \hat{R}_V \hat{S}_V \hat{S}_V \hat{R}_V^T,$$

and correspond with the shape of a triangle $V$. For one face $(\mathbf{v}_1, \mathbf{v}_2, \mathbf{v}_3)$, we define the corresponding Gaussian component:

$$\mathcal{N}((\mathbf{v}_1, \mathbf{v}_2, \mathbf{v}_3)) = \mathcal{N}(\hat{\mathbf{m}}_V, \hat{R}_V, \hat{S}_V).$$

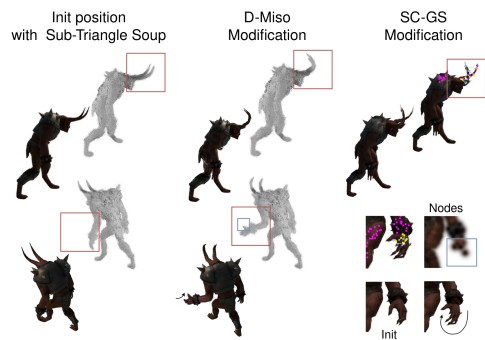

Figure 10: An example of Sub-Triangle Soup modification using D-MiSo and the render obtained by this change from a different viewpoint. It is possible not only to change the position of the hand but also to raise the thumb. Comparison with SC-GS similar modifications, highlighting the challenge of editing small elements individually.

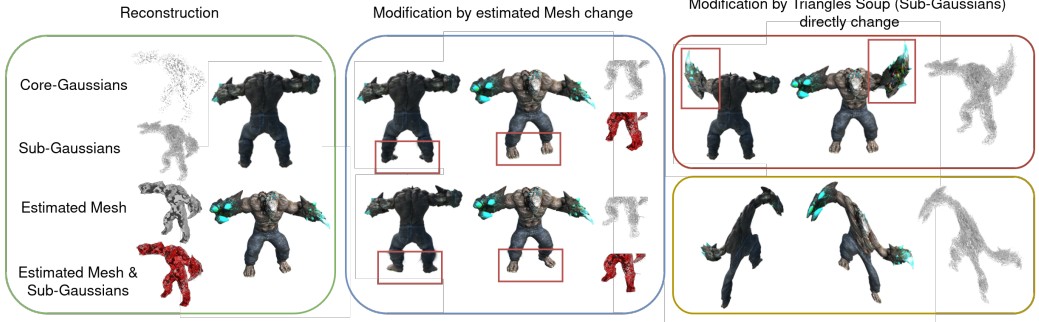

Figure 11: Reconstruction and three ways of modification of the output object. The first involves modifying the estimated mesh, which does not have to be accurate. The next two focus on Sub-Triangle Soup editing. The red box shows the direct modification of the Triangle Soup in a logical way (e.g., raising a hand). The yellow box shows a change in space, i.e., giving fluidity to an object by creating an abstract modification.

Finally, the Gaussian component is derived from the mesh face parameters. This approach can be applied in a Multi-Gaussian framework. Therefore, we will use invertible transformation between Gaussian parameters and triangle face $T$ and notation:

$$\mathcal{N}(V) := \mathcal{N}(\mathcal{T}^{-1}(V)) = \mathcal{N}(\hat{\mathbf{m}}_V, \hat{R}_V, \hat{S}_V).$$

In our D-MiSo we parameterize the $p$ Core-Gaussians with $k$ Triangle soup:

$$\mathcal{G}_{multi} = \left\{ \left( \mathcal{N}_{core}(V_j), \left\{ \left( \mathcal{N}_{sub}\left( \hat{\mathbf{m}}_{V_j} + \hat{R}_{V_j}\boldsymbol{\alpha^i}^T, \hat{R}^i, \hat{S}^i \right), \sigma^i, c^i \right) \right\}_{i=1}^{k} \right) \right\}_{j=1}^{p}, \qquad (3)$$

where $V_j, \boldsymbol{\alpha^i}, R^i, \hat{S}^i, \sigma^i, c^i$ are trainable parameters and $(\hat{\mathbf{m}}_{V_j}, \hat{R}_{V_j}, \hat{S}_{V_j}) = \mathcal{T}^{-1}(V_j)$. Alternatively, we can parameterize Core-Gaussian and Sub-Gaussians by Triangle Soup:

$$\mathcal{G}_{multi} = \left\{ \left( \mathcal{N}_{core}(V_j), \left\{ \left( \mathcal{N}_{sub}\left( V_j^i \right), \sigma^i, c^i \right) \right\}_{i=1}^{k} \right) \right\}_{j=1}^{p}, \qquad (4)$$

where $(\hat{\mathbf{m}}_{V_j}, \hat{R}_{V_j}, \hat{S}_{V_j}) = \mathcal{T}^{-1}(V_j)$, $V^i = \mathcal{T}(\hat{\mathbf{m}}_V + \hat{R}_V\boldsymbol{\alpha^i}^T, \hat{R}^i, \hat{S}^i)$.

In D-MiSo, we use the collation of Multi-Gaussian distribution for rendering and Sub-Triangle Soup for editing. The formal definition of our model uses equation (3) since, in training, we store Core-Gaussian as a triangle face (Core-Gaussian does not have colors) and Sub-Gaussian as a collection from classical GS components with color and opacity. After training, we parametrize our model to equation (4) for editing.

### 3.1   Dynamic Multi-Gaussian Soup (D-MiSo)

Previously, we defined Multi-Gaussians and their parametrization using Triangle Soup. Now, we have all the tools to present the D-MiSo model. The overview of our method is illustrated in Fig. 7. The input to our model is a set of images of a dynamic scene, together with the time label and the corresponding camera poses. Our training is divided into two stages. In the first stage, we initialize the Core-Gaussians. In the second, we add Sub-Gaussian components.

**Stage 1**   First, we train only Core-Gaussians to obtain good initialization for Multi-Gaussins. As Core-Gaussians are mainly employed to capture motion, our model only requires their small amount (Fig. 7). In our approach, the Core-Triangle Soup, constructed via the Core-Gaussians parameterization (as depicted in Fig. 2), is adjusted depending on the time $t$.

In practice, when random initialization of Gaussians is necessary, redundant Gaussians must be pruned first to ensure that the remaining ones represent the object's shape. To reduce the number of Gaussians

and obtain consistent Core-Gaussians, we train GS on a batch containing a few views instead of one. In practice, we render a handful of views from different positions and use back-propagation.

In particular, we parameterize Gaussians $\mathcal{N}(\mathbf{m}, R, S)$ by face $V = (\mathbf{v}_1, \mathbf{v}_2, \mathbf{v}_3)$ to obtain $\mathcal{N}(\mathbf{v}_1, \mathbf{v}_2, \mathbf{v}_3)$. Our *Deform Network* takes as in input the triangle vertices $V$ assigned to the parameterized 3D Core-Gaussians and the current time $t$ and returns updated $\psi(V, t) = (\Delta\mathbf{v}_1(t), \Delta\mathbf{v}_2(t), \Delta\mathbf{v}_3(t), \Delta R_V(t))$. Such updates consist of translation and rotation (Fig. 8):

$$V(t) = V \odot \psi(V, t) = (\mathbf{v}_1 + \Delta\mathbf{v}_1(t), \mathbf{v}_2 + \Delta\mathbf{v}_2(t), \mathbf{v}_3 + \Delta\mathbf{v}_3(t)) \cdot \Delta R_V(t).$$

D-MiSo parameterized Core-Gaussians in time $t$ by: $\mathcal{N}_{core}(V(t), \sigma, c) = \mathcal{N}_{core}(V \odot \psi(V, t), \sigma, c)$ where $\psi$ is a deformable network that moves triangle $V$ according to time $t$. The opacity and color of the Core-Gaussian are used only in the first stage.

**Stage 2: Preparation**   To move to the second phase of the model, it is imperative to prepare the Multi-Gaussians. This involves attaching $k$ Sub-Gaussians to each Core-Gaussian generated in Stage 1, as shown in Fig. 6. Henceforth, Sub-Gaussians assume responsibility for the resultant rendering. Initially, the Sub-Gaussian adopts the same features as the Core-Gaussian, except for the position.

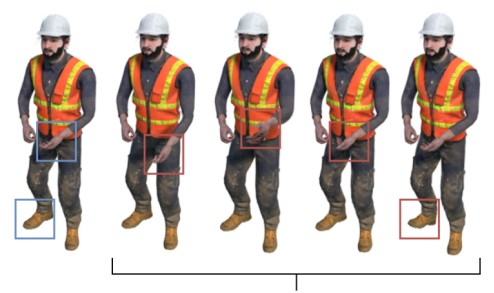

Init position     Small elements animations

**Stage 2**   The primary objective of the second phase is to parallelized Core-Gaussians (Core-Triangles Soup) to enhance understanding of movement and increase rendering quality through the precise training of Sub-Gaussians.

Figure 12: Limbs render and modification obtained with D-MiSo. It is worth noting that it is also possible to close the hand.

Since the centers of Sub-Gaussians are parameterized by the local coordinate system given by the rotation matrix of Core-Gaussian, when the *Deform Network* $\psi$ changes the Core-Gaussian, all Sub-Gaussians (attached to this Core-Gaussian) are modified by global transformation $\psi(V, t)$.

D-MiSo use an additional deformation network *Sub-Rot Network* $\phi$ dedicated to each $i$-th Sub-Gassian's small changes. *Sub-Rot Network* takes the Sub-Gaussian rotation matrix $R_V^i$ and the current time $t$ as input and produces an updated rotation matrix $\Delta R_V^i(t)$.

The position $\hat{\mathbf{m}}_V^i(t) = \hat{\mathbf{m}}_{V(t)} + \hat{R}_{V(t)}\boldsymbol{\alpha}^{\boldsymbol{i}T}$ of the Sub-Gaussian in time $t$ is determined by the position $\hat{\mathbf{m}}_{V(t)}$, and rotation $\hat{R}_{V(t)}$ of the Core-Gaussian (parameterized by triangle $V$) and the learning parameter $\boldsymbol{\alpha}^{\boldsymbol{i}}$. It should be noted that scale $S^i$, color $c_i$, and opacity $\sigma_i$ of Sub-Gaussian are trainable and do not depend on time. *Sub-Rot Network* produce updated $\phi(R^i, t) = \Delta R^i(t)$ for rotation parameter of Sub-Gaussians. Finally parameters of Sub-Gaussians in time $t$ depends on *Deform Network*

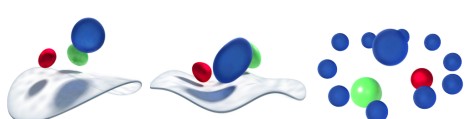

Figure 13: Examples of object modifications. The first method allows for a smooth modification (bending) and also removes (e.g. plate), scales and/or adds (e.g. small blue balls) objects.

$\psi$, and *Sub-Rot Network* $\phi$, and the Corr-Gaussians parameter $V$, and Sub-Gaussian parameters $R^i$, $S^i$, $c_i$ and $\sigma_i$:

$$\mathcal{G}_{Sub}(t) = \{(\mathcal{N}_{sub}(\hat{\mathbf{m}}_{V \odot \psi(V,t)} + \hat{R}_{V \odot \psi(V,t)}\boldsymbol{\alpha}^{\boldsymbol{i}T}, R^i + \phi(R^i, t), S^i), c_i, \sigma_i)\}_{i=1}^k.$$

D-MiSo final model consists of two deformable networks and two levels of Gaussian distributions. This approach enables efficient modeling of object motion over time in dynamic scenes, allowing for adjustments to objects at each time frame.

The result of the model's inference depends on the camera's view angle as well as on the selected time. Only Sub-Gaussian features are used in the rendering process. Hence, the implementation of generating an output image from Gaussians is no different from a vanilla GS.

The modifications involve the appropriate transformation of Sub-Gaussians. We base this on the fact that the transformed Gaussians have exactly the same color properties as before editing. Only the shape of the Gaussians is changed. In our work, we present three editing methods (Fig. 11).

## 4 Experiments

The experimental section is divided into two parts. First, we show that D-MiSo can model dynamic scenes with high quality; in the second part, we demonstrate that our model allows an easy editing procedure, which is our main contribution.

### 4.1 Reconstruction of dynamic scenes

Here, we outline the specifics of our implementation and provide a detailed description of the used datasets. We demonstrated the core advantages of our model by conducting experiments using three different datasets. The source code is available on [3]. Our code is developed on top of the GS vanilla code, according to their license. We used NVIDIA GeForce RTX 4090 and A100 GPUs. The experiments focus on a benchmark task of reconstruction. PSNR metric is used to compare our model with other methods. Furthermore, additional numerical comparisons, i.e., SSIM/LPIPS, are available in the supplementary material.

Table 1: Quantitative comparisons (PSNR) on a D-NeRF dataset showing that D-MiSo gives comparable results with other models.

| | PSNR ↑ | | | | | | |
|---|---|---|---|---|---|---|---|
| | Hook | Jumpin. | Trex | Bounc. | Hell. | Mutant | Stan. |
| D-NeRF [11] | 29.25 | 32.80 | 31.75 | 38.93 | 25.02 | 31.29 | 32.79 |
| TiNeuVox-B [41] | 31.45 | 34.23 | 32.70 | 40.73 | 28.17 | 33.61 | 35.43 |
| Tensor4D [18] | 29.03 | 24.01 | 23.51 | 25.36 | 31.40 | 29.99 | 30.86 |
| K-Planes [17] | 28.59 | 32.27 | 31.41 | 40.61 | 25.27 | 33.79 | 34.31 |
| FF-NVS [42] | 32.29 | 33.55 | 30.71 | 40.02 | 27.71 | 34.97 | 36.91 |
| 4D-GS [28] | 30.99 | 33.59 | 32.16 | 38.59 | 31.39 | 35.98 | 35.37 |
| DynMF [2] | 33.94 | 38.04 | 35.82 | 41.92 | 37.51 | 41.68 | 41.16 |
| Deform-GS [43] | 37.77 | 39.10 | 38.40 | 41.46 | 42.11 | 43.73 | 45.38 |
| Editable | | | | | | | |
| SC-GS [33] | 39.87 | 41.13 | 41.24 | 44.91 | 42.93 | 45.19 | 47.89 |
| **D-MiSo (our)** | 38.13 | 42.05 | 40.88 | 41.49 | 41.49 | 44.38 | 47.66 |

**D-NeRF Datasets:** Contains seven dynamic objects with realistic materials described with a single camera [11]. This means the model had access to only one view at a given moment. Tab. 1 shows we are very comparable to other methods, and we achieve a higher PSNR on one object. The differences in metrics are small, however, our method presents an easier way to edit (Fig. 4). Providing, among other things, better scaling. Following previous methods, our results are obtained on images using 400 by 400 resolution with a black background.

**NeRF-DS[44]:** This dataset contains again seven real-world scenarios containing a moving or deforming specular object. Each scene was recorded using two cameras, with the video captured by one of them being used as a training set, while the footage from the second one was treated as a test set. The camera pose was estimated using COLMAP for both cameras. Tab. 2 shows that our method achieves the SOTA results for distinct objects.

**PanopticSports Datasets:** The dataset comprises six dynamic scenes featuring significant object and actor movements [26, 45]. The scenes are categorized by the activities performed in the video sequences, i.e., juggling, boxing, softball, tennis, football, and basketball. Each scene was recorded using 31 cameras over 150 timesteps. Following the official data split, we use footage from 27 cameras for training and the remaining 4 cameras for testing. Numerical results are shown in Tab. 3. The results show that the model achieved SOTA for five objects according to PSNR metrics and six objects according to LPIPS metrics.

These results demonstrate that D-MiSo is comparable to other methods. Moreover, our main contribution in comparison to other methods is the very easy editing of the resulting object (Fig. 4 11).

### 4.2 Editing of dynamic scenes

As mentioned earlier, we proposed three new methods of output object modification. The first one focuses on moving and preparing a formal mesh, which is connected in contrast to Triangle Soup. We apply a simple meshing strategy on Core-Gaussians. We used the basic Alpha Shape algorithm [46, 47] and showed that the mesh does not have to estimate the surface perfectly to be effective. Then, we reparametrize the Sub-Gaussian by finding the closest face from the mesh (Fig. 9). In practice, we represent each Sub-Gaussian in a local coordinate system (analogically to Multi-Gaussians). Therefore, each Sub-Gaussian is assigned to the nearest face (Fig. 9). Each face can have a different number of Sub-Gaussians prescribed in such a configuration. This method allows us

---

[3] https://github.com/waczjoan/D-MiSo

to maintain the consistency bestowed by the mesh. The whole process is shown in Fig. 5. Thanks to such representation, we can edit our connected mesh to produce the correct edition of the dynamic scene (Fig. 2).

The second editing method allows us to define the connections, i.e., editing Sub-Triangles Soup directly, e.g. moving a hand or bending horns (Fig. 10) or rotating a human body (Fig. 3). Changes are possible because we can also transform a group of Sub-Triangles instead of individual ones. This method allows for even very subtle changes like raising the thumb (Fig. 10), turning the hand over (Fig 4), opening or closing the hand (Fig. 12). These edits would be difficult in other approaches based on adjusting the 3D objects' centroids (nodes) since the space of nodes is limited in details area (Fig. 4). Nodes' use is preeminent for defining movement, which was insignificant in these places.

Table 2: PSNR comparisons on a NeRF-DS dataset showing that D-MiSo gives comparable results with other models.

| | Bell | Sheet | Press | Basin | Cup | Sieve | Plate |
|---|---|---|---|---|---|---|---|
| | | | PSNR ↑ | | | | |
| HyperNeRF [9] | 24.0 | 24.3 | 25.4 | 20.2 | 20.5 | 25.0 | 18.1 |
| NeRF-DS [44] | 23.3 | 25.7 | 26.4 | 20.3 | 24.5 | 26.1 | 20.8 |
| TiNeuVox-B [41] | 23.1 | 21.1 | 24.1 | 20.7 | 20.5 | 20.1 | 20.6 |
| | | | Editable | | | | |
| SC-GS [4] | 25.1 | 26.2 | 26.6 | 19.6 | 24.5 | 26.0 | 20.2 |
| **D-MiSo (our)** | 25.3 | 25.8 | 25.6 | 19.8 | 24.5 | 26.5 | 20.8 |

With this method, we can also change complicated objects like a $360°$ scene without losing the dynamics-related model. For example, the rotation of a person stacking boxes (Fig. 3).

Table 3: Comparison on PanopticSports dataset.

| Metrics | Method | Juggle | Boxes | Softball | Tennis | Football | Basketball | Mean |
|---|---|---|---|---|---|---|---|---|
| PSNR ↑ | 3DGS [1] | 28.19 | 28.74 | 28.77 | 28.03 | 28.49 | 27.02 | 28.21 |
| | Dyn3DG [26] | 29.48 | 29.46 | 28.43 | 28.11 | 28.49 | 28.22 | 28.7 |
| | D-MiSo (our) | 29.79 | 29.39 | 28.6 | 29.02 | 28.99 | 28.49 | 29.04 |
| SSIM ↑ | 3DGS [1] | 0.91 | 0.91 | 0.91 | 0.90 | 0.90 | 0.89 | 0.90 |
| | Dyn3DG [26] | 0.92 | 0.91 | 0.91 | 0.91 | 0.91 | 0.91 | 0.91 |
| | D-MiSo (our) | 0.93 | 0.92 | 0.92 | 0.92 | 0.92 | 0.91 | 0.92 |
| LPIPS ↓ | 3DGS [1] | 0.15 | 0.15 | 0.14 | 0.16 | 0.16 | 0.18 | 0.16 |
| | Dyn3DG [26] | 0.15 | 0.17 | 0.19 | 0.17 | 0.19 | 0.18 | 0.17 |
| | D-MiSo (our) | 0.13 | 0.13 | 0.15 | 0.14 | 0.13 | 0.15 | 0.14 |

The third method focuses on transforming the space in which the object is located. Similar to the second method, the third one also works directly on the Sub-Triangle Soup. We can achieve such an effect by applying a certain function to the selected plane. In practice, the object, or a portion of it, is modified, for instance, through the use of sinusoidal functions. It allows us to obtain fluidity and more easily define the physical nature of the movement (Fig. 1, 11, 13).

Our methods are also scalable, so we can easily remove or duplicate elements from an image. Moreover, the duplicated elements can be given their own dynamics. Examples of these effects are shown in Fig. 13, where we removed the plate and both duplicated and rescaled the blue balls multiple times.

## 5 Conclusion

D-MiSo is a novel method based on Gaussian Splatting parameterization, which produces a cloud of triangles called Triangle Soup. The method allows easy editing of objects created in inference with the possible transformations, including moving, scaling, and rotating. By defining Multi-Gaussians, the obligatory separability of modified elements seen in other models is combated. In addition, certain elements of objects can be duplicated and removed (Fig. 13). Furthermore, the D-MiSo method can facilitate giving different dynamics to separate parts of an object (Fig. 1).

**Limitation** The method allows for complex changes at a given moment in time. However, if some area is not well represented in the training set, it is impossible to edit them. For example, a person's hand can be changed but not fingers (Fig. 4). This is due to the liminality of Triangle Soup relative to a well-fitted mesh.

**Broader impact** Our model significantly improves rendering quality and advances 3D scene reconstruction and rendering, impacting multiple domains by enabling more realistic and efficient 3D modeling and animation. This technology could enhance VR/AR experiences [48], robotics [49], and medical imaging [50]. It could also be used for interactive education [50], scientific visualization, and a plethora of other commercial applications like product design and real estate [51].

**Acknowledgments** The work of P. Borycki was supported by the National Centre of Science (Poland) Grant No. 2021/41/B/ST6/01370. The work of J. Kaleta was supported by the National Science Centre, Poland, grant no 2022/47/O/ST6/01407. The work of P. Spurek and J. Waczyńska was supported by the National Centre of Science, Poland Grant No. 2021/43/B/ST6/01456.

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

# A Extended related work and potential applications of D-MiSo

The detail-capturing capabilities of Sub-Gaussians can be effectively adapted to other GS-based approaches, particularly in cases where native methods lack dedicated support for high-detail reconstruction [52, 53, 54] or edition [55, 56].

Recent studies have investigated the integration of human faces, avatars, and meshes for 3D human modeling and rendering from video, presenting a promising avenue for enhancing detailed reconstruction.

For instance, [57] introduces a human body representation where neural features across frames share a latent code anchored to a deformable mesh, facilitating cross-frame integration and efficient 3D representation learning in sparse monocular video scenarios. Similarly, HumanNeRF [58] optimizes a volumetric human representation in a canonical T-pose, using a decomposed motion field to handle both skeletal and non-rigid deformations. This method provides high-quality, free-viewpoint renderings in challenging settings. UV-Volumes [59] further address the high computational costs associated with NeRF, achieving real-time, editable video rendering by encoding appearance details into 2D neural texture stacks for efficient 3D density and texture estimation.

Gaussian-based representations have gained traction for modeling dynamic avatars, as in [60, 61, 62, 63, 64], with most approaches leveraging video data and human pose/face priors like FLAME fitting [65]. Building on these advances, we propose that D-MiSo concepts could be effectively adapted for dynamic motion modeling and small detail capture in avatar representations. Specifically, Stage 1 could focus on integrating such models for capturing fine-grained avatar motion with anchored details.

# B Extension of numerical results from the main paper

This section contains a numerical comparison of the D-MiSo with other models regarding the experiments described in the main document. We used additional LPIPS and SSIM metrics. During the experiments, we used the RTX4090 GPU and densification until 5000 iterations (1st Stage). We see that we get comparable competitive results. We can see that our method achieves better results for five objects from the D-NeRF dataset in terms of the LPIPS metric (Tab. 4). Full numerical comparison with metrics for the NeRF-DS dataset shows that we are comparable to other methods in the reproduction task (Tab. 5).

Table 4: Quantitative comparison on a D-NeRF dataset

| Methods | Hook | | | Jumpingjacks | | | Trex | | | BouncingBalls | | |
|---|---|---|---|---|---|---|---|---|---|---|---|---|
| | PSNR ↑ | SSIM ↑ | LPIPS ↓ | PSNR ↑ | SSIM ↑ | LPIPS ↓ | PSNR ↑ | SSIM ↑ | LPIPS ↓ | PSNR ↑ | SSIM ↑ | LPIPS ↓ |
| D-NeRF | 29.25 | .968 | .1120 | 32.80 | .981 | .0381 | 31.75 | .974 | .0367 | 38.93 | .987 | .1074 |
| TiNeuVox-B | 31.45 | .971 | .0569 | 34.23 | .986 | .0383 | 32.70 | .987 | .0340 | 40.73 | .991 | .0472 |
| Tensor4D | 29.03 | .955 | .0499 | 24.01 | .919 | .0768 | 23.51 | .934 | .0640 | 25.36 | .961 | .0411 |
| K-Planes | 28.59 | .953 | .0581 | 32.27 | .971 | .0389 | 31.41 | .980 | .0234 | 40.61 | .991 | .0297 |
| FF-NVS | 32.29 | .980 | .0400 | 33.55 | .980 | .0300 | 30.71 | .960 | .0400 | 40.02 | .990 | .0400 |
| 4D-GS | 30.99 | .990 | .0248 | 33.59 | .990 | .0242 | 32.16 | .988 | .0216 | 38.59 | .993 | .0267 |
| SC-GS | 39.87 | .997 | .0076 | 41.13 | .998 | .0067 | 41.24 | .998 | .0046 | 44.91 | .998 | .0166 |
| D-MiSo (our) | 38.13 | .990 | .0086 | 42.05 | .995 | .0049 | 40.88 | .996 | .0029 | 41.49 | .993 | .0079 |

| Methods | Hellwarrior | | | Mutant | | | Standup | | | Average | | |
|---|---|---|---|---|---|---|---|---|---|---|---|---|
| | PSNR ↑ | SSIM ↑ | LPIPS ↓ | PSNR ↑ | SSIM ↑ | LPIPS ↓ | PSNR ↑ | SSIM ↑ | LPIPS ↓ | PSNR ↑ | SSIM ↑ | LPIPS ↓ |
| D-NeRF | 25.02 | .955 | .0633 | 31.29 | .978 | .0212 | 32.79 | .991 | .0241 | 31.69 | .975 | .0575 |
| TiNeuVox-B | 28.17 | .978 | .0706 | 33.61 | .982 | .0388 | 35.43 | .991 | .0230 | 33.76 | .983 | .0441 |
| Tensor4D | 31.40 | .925 | .0675 | 29.99 | .951 | .0422 | 30.86 | .964 | .0214 | 27.62 | .947 | .0471 |
| K-Planes | 25.27 | .948 | .0775 | 33.79 | .982 | .0207 | 34.31 | .984 | .0194 | 32.32 | .973 | .0382 |
| FF-NVS | 27.71 | .970 | .0500 | 34.97 | .980 | .0300 | 36.91 | .990 | .0200 | 33.73 | .979 | .0357 |
| 4D-GS | 31.39 | .974 | .0436 | 35.98 | .996 | .0120 | 35.37 | .994 | .0136 | 34.01 | .987 | .0316 |
| SC-GS | 42.93 | .994 | .0155 | 45.19 | .999 | .0028 | 47.89 | .999 | .0023 | 43.31 | .997 | .0063 |
| D-MiSo (our) | 41.49 | .986 | .0173 | 44.38 | .997 | .0026 | 47.66 | .998 | .0016 | 42.27 | .993 | .0065 |

Table 5: Quantitative comparison on a NeRF-DS dataset

| Methods | Bell | | | Sheet | | | Press | | | Basin | | |
|---|---|---|---|---|---|---|---|---|---|---|---|---|
| | PSNR ↑ | SSIM ↑ | LPIPS ↓ | PSNR ↑ | SSIM ↑ | LPIPS ↓ | PSNR ↑ | SSIM ↑ | LPIPS ↓ | PSNR ↑ | SSIM ↑ | LPIPS ↓ |
| HyperNeRF | 24.0 | .884 | .159 | 24.3 | .874 | .148 | 25.4 | .873 | .164 | 20.2 | .829 | .168 |
| NeRF-DS | 23.3 | .872 | .134 | 25.7 | .918 | .115 | 26.4 | .911 | .123 | 20.3 | .868 | .127 |
| TiNeuVox-B | 23.1 | .876 | .113 | 21.1 | .745 | .234 | 24.1 | .892 | .133 | 20.7 | .896 | .105 |
| SC-GS | 25.1 | .918 | .117 | 26.2 | .898 | .142 | 26.6 | .901 | .135 | 19.6 | .846 | .154 |
| D-MiSo (our) | 25.3 | .846 | .174 | 25.8 | .875 | .211 | 25.6 | .867 | .206 | 19.8 | .788 | .217 |

| Methods | Cup | | | Sieve | | | Plant | | | Average | | |
|---|---|---|---|---|---|---|---|---|---|---|---|---|
| | PSNR ↑ | SSIM ↑ | LPIPS ↓ | PSNR ↑ | SSIM ↑ | LPIPS ↓ | PSNR ↑ | SSIM ↑ | LPIPS ↓ | PSNR ↑ | SSIM ↑ | LPIPS ↓ |
| HyperNeRF | 20.5 | .705 | .318 | 25.0 | .909 | .129 | 18.1 | .714 | .359 | 22.5 | .827 | .206 |
| NeRF-DS | 24.5 | .916 | .118 | 26.1 | .935 | .108 | 20.8 | .867 | .164 | 23.9 | .898 | .127 |
| TiNeuVox-B | 20.5 | .806 | .182 | 20.1 | .822 | .205 | 20.6 | .863 | .161 | 21.5 | .843 | .162 |
| SC-GS | 24.5 | .916 | .115 | 26.0 | .919 | .114 | 20.2 | .837 | .202 | 24.1 | .891 | .140 |
| D-MiSo (our) | 24.5 | .874 | .185 | 26.5 | .881 | .159 | 20.8 | .815 | .234 | 24.0 | .849 | .198 |

Table 6: Storage cost, training time, fps, and batch study on a D-NeRF dataset

| | PSNR ↑ | | | | | | |
|---|---|---|---|---|---|---|---|
| | Hook | Jumpin. | Trex | Bounc. | Hell. | Mutant | Standup |
| Batch: | 4 | 4 | 4 | 4 | 4 | 4 | 4 |
| PSNR | 37.77 | 40.42 | 39.56 | 40.63 | 41.44 | 43.38 | 46.07 |
| Time 1st stage | 00:02:48 | 00:02:22 | 00:02:41 | 00:03:12 | 00:02:27 | 00:03:46 | 00:03:17 |
| Time 2nd stage | 1:40:15 | 1:16:18 | 2:14:09 | 1:43:08 | 1:01:27 | 1:48:49 | 1:19:32 |
| Storage cost | 76MB | 53.5MB | 122MB | 131MB | 27MB | 73MB | 43MB |
| Rendering time [fps] | 138 | 175 | 90 | 123 | 185 | 138 | 169 |
| Batch: | 8 | 8 | 8 | 8 | 8 | 8 | 8 |
| PSNR | 38.07 | 41.65 | 40.74 | 40.55 | 41.59 | 44.40 | 47.22 |
| Time 1st stage [h] | 00:05:30 | 0:05:17 | 00:05:38 | 00:05:16 | 00:04:28 | 00:05:51 | 00:05:56 |
| Time 2nd stage [h] | 2:27:08 | 2:03:25 | 2:42:22 | 2:40:30 | 1:43:33 | 2:18:43 | 2:12:12 |
| Storage cost | 32MB | 24MB | 51MB | 80MB | 16MB | 28MB | 20MB |
| Rendering time[fps] | 192 | 190 | 143 | 153 | 205 | 188 | 194 |

In our method, a batch of images is taken as input to the model. Tab. 6 shows a numerical comparison using batch: 4, 8 on D-NeRF. Training takes 80 thousand iterations, and the second stage starts at the 5 thousandth iteration. Each Core-Gaussian has attached 25 Sub-Gaussians. This is comparable to the SC-GS implementation. Tab. 6 also presents the training time and storage cost, as well as the FPS for rendering for each dataset. These results suggest that our model is memory efficient and that the time required to train it is minimal. With batch equal to 8 in all cases, storage costs decreased, and performance improved with a trade-off of increased training time.

A similar batch study has been done for the NeRF-DS dataset presented in Tab.7. We can see that batch is playing a bigger role in the dataset, which is considered a more pronounced move.

We contrasted our model with and without the Sub-Rotation Network. The Sub-Rot Network, despite being a simple single-layer network, plays a crucial role in the second stage. Tab. 8 shows a numerical comparison of PSNR for models with and without the Sub-Rot Network. Furthermore, we would like to emphasize that due to its shallow architecture, incorporating the Sub-Rot Network does not significantly impact training time. We decided to present the results on the *jumpingjack* dataset distinguishing between batches of 4, 8. These results show that the Sub-Rot Network is crucial to obtaining SOTA PSNR. With Sub-Rot Network, we obtain approximately 0.5 PSNR crucial to obtain the reconstruction of small elements (like human fingers in the *jumpingjack* dataset)

Tab. 9 shows how the size of the network influences training time and render speed. These results show that deformable networks and Sub-Rot Networks are not costly in terms of rendering time, resulting in real-time rendering. In practice, batch size has a higher impact than deformation network depth.

Table 7: Batch study on a NeRF-DS dataset

| Batch size | Bell | | | Sheet | | | Press | | | Basin | | |
|---|---|---|---|---|---|---|---|---|---|---|---|---|
| | PSNR ↑ | SSIM ↑ | LPIPS ↓ | PSNR ↑ | SSIM ↑ | LPIPS ↓ | PSNR ↑ | SSIM ↑ | LPIPS ↓ | PSNR ↑ | SSIM ↑ | LPIPS ↓ |
| 8 | 25.1 | .851 | .165 | 24.8 | .863 | .219 | 25.2 | .858 | .205 | 19.7 | .788 | .217 |
| 4 | 24.5 | .832 | .190 | 25.1 | .873 | .202 | 25.3 | .860 | .215 | 19.8 | .781 | .245 |
| 2 | 25.3 | .846 | .174 | 25.8 | .875 | .211 | 25.6 | .867 | .206 | 19.6 | .785 | .207 |
| 1 | 25.1 | .844 | .190 | 25.6 | .873 | .216 | 24.3 | .842 | .273 | 19.7 | .785 | .229 |
| Batch size | Cup | | | Sieve | | | Plant | | | Average | | |
| | PSNR ↑ | SSIM ↑ | LPIPS ↓ | PSNR ↑ | SSIM ↑ | LPIPS ↓ | PSNR ↑ | SSIM ↑ | LPIPS ↓ | PSNR ↑ | SSIM ↑ | LPIPS ↓ |
| 8 | 24.0 | .884 | .163 | 26.5 | .881 | .158 | 20.2 | .812 | .226 | 23.6 | .848 | .193 |
| 4 | 24.2 | .886 | .163 | 25.8 | .871 | .167 | 20.8 | .815 | .234 | 23.6 | .845 | .202 |
| 2 | 24.4 | .883 | .169 | 26.0 | .875 | .169 | 20.6 | .817 | .232 | 23.9 | .849 | .195 |
| 1 | 24.5 | .874 | .185 | 25.8 | .864 | .195 | 20.5 | .808 | .245 | 23.6 | .841 | .219 |

Our model operates on several hyperparameters, primarily derived from the basic GS framework. Given the introduction of additional stages and the multi-Gaussian component, one of the new hyperparameters is the number of Core-Gaussians and Sub-Gaussians (Tab. 10). The ablation study also focuses on the influence of the Sub-Rot Network on training time and PSNR (Tab. 8). Tables show training time and PSNR analysis.

Table 8: Training time corresponding to the number of Sub-Gaussians and iteration start of the deform network using the *jumpingjack* dataset.

| | batch = 4 | | batch = 8 | |
|---|---|---|---|---|
| Sub-Rot Network | with | without | with | without |
| PSNR | 40.42 | 39.98 | 41.65 | 41.27 |
| Training time [h] | 1:18 | 1:12 | 2:08 | 1:45 |
| Rendering time [fps] | 175 | 186 | 190 | 259 |

The number of Core-Gaussians is determined automatically during stage 1, utilizing the pruning mechanism implemented in GS. The Tab. 10 illustrates the training time corresponding to these parameters using the *jumpingjacks* dataset as an example with batch equal to 8. We can see the speed of the 1st stage, which is used for Core-Gaussian preparation. Moreover, too few Core-Gaussians can cause a drop in render quality (PSNR metric). The experiments suggest the number of Core-Gaussians is sufficient after the first phase. In all cases, a larger number of Sub-Gaussians improved the quality of renders.

Table 9: Deformation network depth (numbers of layers) and batch study on a *jumpingjack* dataset from D-NeRF.

| Def. net. depth | 4 | 6 | 8 | 10 | 4 | 6 | 8 | 10 |
|---|---|---|---|---|---|---|---|---|
| Batch | 4 | 4 | 4 | 4 | 8 | 8 | 8 | 8 |
| PSNR | 40.91 | 40.75 | 40.42 | 40.26 | 41.86 | 42.01 | 41.65 | 41.37 |
| Training time [h] | 1:21 | 1:22 | 1:18 | 1:31 | 2:00 | 1:58 | 2:08 | 2:07 |
| Render time [fps] | 138 | 167 | 175 | 144 | 227 | 221 | 190 | 192 |

Table 10: Training time corresponding to the number of Sub-Gaussians and iteration start of the deform network using the *jumpingjack* dataset.

| n_sub | 1 | 10 | 25 | 1 | 10 | 25 | 1 | 10 | 25 |
|---|---|---|---|---|---|---|---|---|---|
| Iter. start of def. net. | 2000 | 2000 | 2000 | 3000 | 3000 | 3000 | 5000 | 5000 | 5000 |
| N_core after 1st stage | 3290 | 3340 | 3434 | 2841 | 2974 | 2936 | 1803 | 1798 | 1807 |
| PSNR | 38.94 | 41.35 | 41.65 | 38.52 | 41.14 | 41.41 | 33.63 | 37.14 | 37.47 |
| Train time 1st stage [h] | 00:05:14 | 00:04:48 | 00:05:17 | 00:04:17 | 00:04:33 | 00:04:16 | 00:03:27 | 00:03:02 | 00:03:21 |
| Train time 2nd stage [h] | 1:51:17 | 1:44:21 | 2:03:25 | 1:39:39 | 1:53:18 | 1:50:25 | 1:48:54 | 1:41:12 | 1:56:28 |
| Sum: Train time [h] | 1:56:31 | 1:49:09 | 2:08:42 | 1:43:56 | 1:57:51 | 1:54:41 | 1:52:21 | 1:44:14 | 1:59:49 |

## C  Extension of examples modification

Below is an example of scene modification from the PanopticSports dataset (Fig. 14). The dynamics of the object were stopped, so as to show the possibility of changing the position of the ball at a given time instant $t$.

Due to the characteristics of Multi-Gaussians and the ease of their control, we show that we can easily duplicate and/or scale selected objects (Fig. 15).

The third method of modification allows us to give new dynamics and fluidity to the object (Fig. 16). We can see that even difficult edits like bending the back, changing the shape of the face, and bending the hand are possible while the result is natural from the point of view of graphics. This is made possible by editing the scale and rotation of Gaussians in an explicitly defined way.

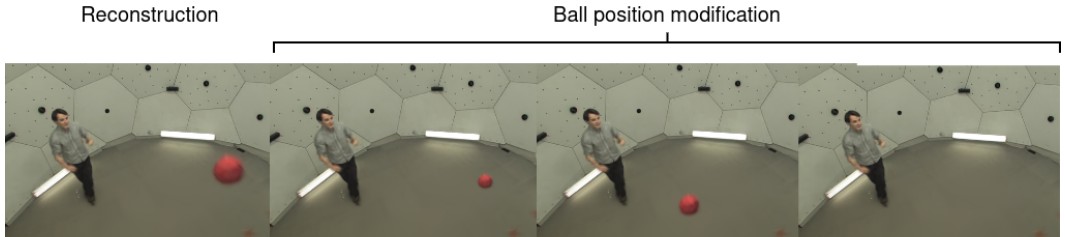

Figure 14: Example of a ball position modification on a $360°$ scene from PanopticSports Datasets

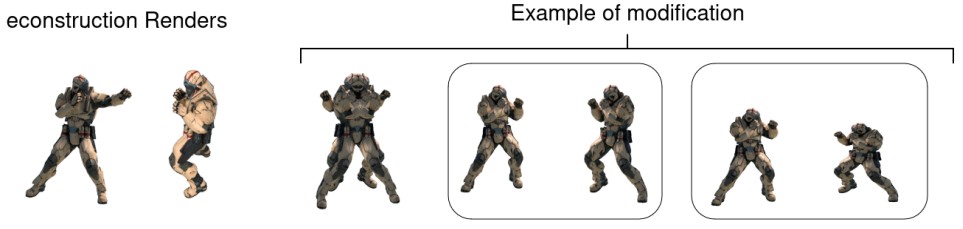

Figure 15: Example of animation by duplicating elements or changing their scale

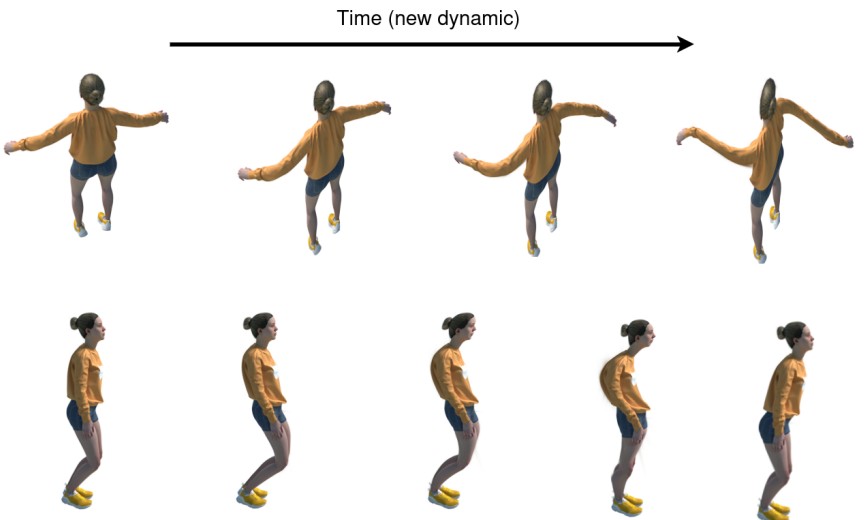

Figure 16: Example of a third way of modification: transformation of space, allowing to give new dynamics.

