# OpenReview forum: "D-MiSo: Editing Dynamic 3D Scenes using Multi-Gaussians Soup"
_NeurIPS.cc/2024/Conference — NeurIPS 2024 poster_

### Official Review · Reviewer_ZSU4 · 2024-06-16

**Soundness:** 3
**Presentation:** 2
**Contribution:** 2
**Rating:** 5
**Confidence:** 4

**Summary:**

This paper proposes the D-Miso pipeline, which includes multi-Gaussian components and two deformation networks to model global transformation and local deformation. A good training strategy is introduced to fit the proposed methods. The rendering quality looks good, and the editing results seem better than those of current motion-based sparse control point deformation using 3D Gaussians.

**Strengths:**

1. The paper proposes the multi-Gaussian component, allowing large 3D scene elements. It seems like a novel idea.
2. D-Miso uses multi-Gaussian components and two deformation networks for modeling dynamic scenes.
3. The editing results seem better than those of SC-GS.

**Weaknesses:**

1. Multi-Gaussian representations are similar to SC-GS's control points. Since 3D/4D Gaussians themselves can model large 3D scene elements, this is not a strength. I think the contribution is minor, and it seems that no ablation study is presented in the paper.
2. Most demos are provided in synthetic datasets, which is not convincing. Results on real-world datasets such as [9, 13] are recommended to be included.
3.The "Related Works" section should be reorganized. More discussions on multi-view dynamic scenes and flow-based radiance fields should be included.
4. If your method is based on [3], a brief review should be included in the main text as a preliminary to improve the presentation.
5. More details such as storage costs and training time (for each stage) should be included. Additionally, would the rendering speed be affected by the large deformation network?
6. Minor issues: Line 140: "Guassinas" -> "Gaussians," Line 197: "vertixes" -> "vertices," Fig. 10: "possition" should be "position."
7. What is n_core as mentioned in Eq.1?

**Questions:**

1. More comparison and discussions with SC-GS should be included since SC-GS can also support editing and share the similar ideas with control points/Gaussians.
2. As I understand, deformation-based methods find it challenging to synthesize correct novel views with large motion. Could the authors provide comprehensive insight into this issue?

**Limitations:**

Yes, the limiations have been addressed.

---

> ### Author Rebuttal · Authors · 2024-08-05
>
> We thank the reviewer for the remarks and excellent comments and appreciate the time taken to review our manuscript. We have responded to these remarks below.
>
> ***W1 Multi-Gaussian representations are similar to SC-GS's control points ...***
>
> Our solution differs in architecture and training strategy from SC-GS, which uses control points while we employ two hierarchies of Gaussian components. **More significantly, we have an extremely large possibility of editing dynamic scenes.** All other Reviewers agree that D-Miso introduces significant novelty, as underscored by Reviewer 2 (“Experimental results show that this method not only matches the rendering quality of SC-GS but **also enables the editing of more extreme large motions**”). The SC-GS authors use control points, which influence a large part of the object. Therefore, when we modify some object element (e.g., the hand of the object, see Fig.4 in our manuscript), the full object changes (in the case of the example from Fig. 4, the head is moving). Thanks to using two levels of Gaussian components, we can modify objects easily without artefacts (see Fig. 4 in the main paper). **Consequently, our new GS representation is better suited for editing dynamic scenes than SC-GS, and it is the paper's main contribution.**
>
> ***W2A Most demos are provided in synthetic datasets, which is not convincing. Results on real-world datasets such as [9, 13] are recommended to be included.***
>
> The SC-GS and other baseline models, which modify dynamic scenes, use only synthetic examples. In contrast, **our paper includes examples of full scenes (please see Fig. 3)**, where we modified real-world dynamic scenes. Furthermore, in  Appendix C and Fig. 14, we will also provide additional real-scenes modifications. We also add movies (GIFs) from real scene modification to supplementary materials. We agree that such experiments are valuable, and **we will move more examples from the appendix to the main paper.**
>
> ***W2B The "Related Works" section should be reorganized. More discussions on multi-view dynamic scenes and flow-based radiance fields should be included.***
>
> Early works on dynamic scenes face difficulties when dealing with monocular settings and uncontrolled or lengthy scenarios. To enhance scene motion modeling, some works in **NeRFs utilize flow-based techniques** [10, g1, g2]. Method [10] extends the static scenario by including time in the domain and explicitly modeling 3D motion as dense scene flow fields. For a given 3D point and time, the model predicts both reflectance and opacity and forward and backward 3D scene flow. In [g1], the model learns scene appearance, density, and motion jointly. NeRFlow internally represents scene appearance and density with a neural radiance field and scene dynamics with a flow field. Finally, [g2] focuses on predicting explicit 3D correspondence between neighboring frames to achieve information aggregation. The correspondence prediction is achieved through the estimation of consistent depth and scene flow information across frames with the use of dedicated networks.
>
> Following suggestions from another Reviewer (**hxyh**; W1), we plan to include **additional works that operate in dynamic scene reconstruction:** [b1, b2]. These are early Gaussian splatting works that significantly contributed to the dynamic scene reconstruction field, and we plan to include them in the revised version of the manuscript. These methods, however, require a multi-view setup similar to those already included [21].
>
> [g1] Yilu et al. 2021, “Neural radiance flow for 4D view synthesis and video processing,” in Proceedings of the IEEE/CVF ICCV
>
> [g2] Zhou et al. 2024, "DynPoint: Dynamic Neural Point For View Synthesis," in NeurIPS 36
>
> ***W3 “If your method is based on [3], a brief review should be included in the main text as a preliminary to improve the presentation.”***
>
> Our model is based upon the Gaussian parameterization framework outlined in [3], with a detailed elaboration provided in Appendix A. To enhance readability, this section will be moved to the main text.
>
> ***W4 More details such as storage costs and training time (for each stage) should be included. Additionally, would the rendering speed be affected by the large deformation network?”***
>
> Due to the rebuttal length, ablation studies are included in the global rebuttal.
>
> **W6 “What is n_core as mentioned in Eq.1?”**
>
> We have introduced the notation N_Core​ (Eq. 1) and N_Sub​ (Eq. 3) to clarify our references to Gaussian distributions for Core-Gaussians versus Sub-Gaussians.
>
> ***Q1  “More comparison and discussions with SC-GS should be included since SC-GS can also support editing and share similar ideas with control points/Gaussians.”***
>
> We provide numerical comparisons with SC-GS in the main paper body (Tab 1, Tab 2, Appendix: Table 5 and Table 6 in our draft paper). Fig. 4 presents a visual comparison with the SC-GS model, highlighting the placement of key points in SC-GS. The SC-GS model lacks key points necessary for hand rotation, as demonstrated in our method (third image, second column). Additionally, overextending the SC-GS model results in "tearing" the human's arm, whereas our method handles scaling more effectively. Additionally, we will add more visual comparisons to illustrate our approach's robustness clearly.
>
> ***Q2  “As I understand, deformation-based methods find it challenging to synthesize correct novel views with large motion. Could the authors provide comprehensive insight into this issue?“***
>
> Models based on deformable networks, including our model and SC-GS, create representations of key components (such as Core-Gaussians in our case and landmarks points in SC-GS) by positioning them optimally for ease of adjustment at each time instant. However, when there is significant movement in the training set, the network responsible for component localization may face challenges. Such a problem is the same in all models which use deformable networks.

---

> > ### Comment · Reviewer_ZSU4 · 2024-08-08
> >
> > Thanks for your comprehensive answers, most of my questions are solved and would like to change my score to BA.

---

> > > ### Author Response · Authors · 2024-08-08
> > > **Authors’ response**
> > >
> > > We thank the Reviewer for the constructive feedback and appreciate increasing the score.

---

### Official Review · Reviewer_qmWF · 2024-06-22

**Soundness:** 3
**Presentation:** 2
**Contribution:** 3
**Rating:** 6
**Confidence:** 2

**Summary:**

This paper is developed based on SC-GS, which enhanced 3DGS with Deformed Control Points to model low-rank dynamics and modify dynamic objects over time. SC-GS necessitates selecting elements that need to be kept fixed and centroids that should be adjusted throughout editing, and it poses additional difficulties regarding re-productivity editing. This paper proposes Dynamic Multi-Gaussian Soup (D-MiSo), which serves as a kind of mesh representation of dynamic GS, links parameterized Gaussian splats, forming a Triangle Soup with the estimated mesh, and separately constructs new trajectories for the 3D objects composing the scene, which makes the scene’s dynamic editable over time or while maintaining partial dynamics.

**Strengths:**

D-MiSo estimates the mesh as a set of disconnected triangle faces in Multi-Gaussians and uses a dynamic function to control the vertices, which makes dynamic 3D-GS easier to modify than SC-GS.

Multi-Gaussians = Core-Gaussians + Sub-Gaussians. Core-Gaussians are an alternative to the control points discussed in SC-GS, with the added advantage of allowing individual modifications. Sub-Gaussians are defined by principal components of core-Gaussian, such that modifying Core-Gaussians will also change Them, which allows scene modifications.

Although the positions of Core-Gaussians are learned by deformation MLP during training, this paper can modify dynamic objects by using the vertex of the Sub-Gaussians or generate mesh from the Core-Gaussians during inference.

D-MiSo can also handle affine transformation and object scaling.

**Weaknesses:**

Lack of training time comparisons.

Lack of comparison against gs2mesh-based methods and cage-based methods.

**Questions:**

Lack of discussion with respect to the human-NeRF (Neural Body/HumanNeRF/UV-Volumes) and human-GS methods (Animatable Gaussians/D3GA/GoMAvatar).

**Limitations:**

The authors adequately addressed the limitation.

---

> ### Author Rebuttal · Authors · 2024-08-05
>
> We thank the Reviewer for the feedback and for pointing out improvements to our paper. We respond to these concerns in the points below.
>
> ***W1: Lack of training time comparisons.***
>
>
> Here, we showcase a time comparison between SC-GS and D-Miso models, with the former operating noticeably faster than the latter. This can be attributed to two causes. First, unlike SC-GS, D-Miso benefits significantly from a larger batch size during training. Second, D-Miso incorporates two layers of Gaussian components (Core-Gaussians and Sub-Gaussians). On the other hand, SC-GS uses landmark points, making it faster in training but more constrained in editing and processing full scenes. Furthermore, our model obtains better results on real scenes, see Tab. 3 and Tab. 4.
>
>
> Dataset:| Hook|Jumpin|Trex|Bounce|Hell|Mutant|Standup|
> |---|---|---|---|---|---|---|---|
> | | | |**D-Miso**| | | | |
> Batch:|1|1|1|1|1|1|1|
> Time 1st stage [h] | 0:00:53 | 0:00:56 | 0:01:15 | 0:01:21 | 0:00:46 | 0:01:10 | 0:00:56|
> Time 2nd stage [h] | 0:59:03|0:53:41|1:13:51|1:23:45|0:24:35|1:09:26|0:41:41|
> Batch:|4|4|4|4|4|4|4|
> Time 1st stage [h] |00:02:48|00:02:22|00:02:41|00:03:12|00:02:27|00:03:46|00:03:17
> Time 2nd stage  [h] |1:40:15|1:16:18|2:14:09|1:43:08|1:01:27|1:48:49|1:19:32
> Batch:|8|8|8|8|8|8|8|
> |Time 1st stage [h] |00:05:30|0:05:17|00:05:38|00:05:16|00:04:28|00:05:51|00:05:56|
> |Time 2nd stage [h] |2:27:08|2:03:25|2:42:22|2:40:30|1:43:33|2:18:43|2:12:12|
> | | | |**SC-GS**| | | | |
> Batch:|1|1|1|1|1|1|1|
> |Time|0:24:52|0:21:31|0:30:05|0:30:02|0:17:37|0:27:26|0:20:23|
>
>
>
>
> ***W2: Lack of comparison against gs2mesh-based methods and cage-based methods.***
>
> **- GS2Mesh based methods:**
> We have already included the following mesh-based Gaussian splatting works, which operate in static setups [3, 26, 27].
> To address the Reviewer’s remark, we will also include a more recent work that combines meshes with Gaussians and operates in dynamic scenarios [e1]. This framework reconstructs a high-fidelity and time-consistent mesh from a single monocular video. Building on this representation, DG-Mesh recovers high-quality meshes from the Gaussian points and can track mesh vertices over time, enabling applications such as texture editing on dynamic objects. However, the method relies on a Poisson Solver and differentiable Marching Cubes to recover the deformed surface, significantly complicating and slowing down the pipeline. Moreover, it does not explore geometry modification capabilities, which constitute a significant aspect of our work.
>
> **- Cage based methods:**
> We thank the reviewer for suggestions regarding cage-based methods. In response, we will include two recent works that utilize the cage concept. Cage-based methods like [e2, e3] are crucial for the efficient and intuitive manipulation of 3D scenes. By providing a direct deformation approach, these methods simplify a traditionally complex task, making it more accessible. However, they require complex cage-building steps, which complicates the pipeline. These works in their current form do not address training on dynamic scenes so direct comparison to our method is not feasible. Additionally, cage-based methods may lack the flexibility and precision of manual, vertex-level deformation techniques, potentially missing fine details. In contrast, our work emphasizes small and precise element movement, addressing this limitation.
>
> [e1] Liu et al. 2024, “Dynamic Gaussians Mesh: Consistent Mesh Reconstruction from Monocular Videos,” arXiv preprint arXiv:2404.12379. Retrieved from https://arxiv.org/abs/2404.12379
>
> [e2] Huang et al. 2024, “GSDeformer: Direct Cage-based Deformation for 3D Gaussian Splatting,” arXiv preprint arXiv:2405.15491. Retrieved from https://arxiv.org/abs/2405.15491
>
> [e3] Jiang et al. 2024, “VR-GS: A Physical Dynamics-Aware Interactive Gaussian Splatting System in Virtual Reality,” arXiv preprint arXiv:2401.16663
>
> ***Q1: Lack of discussion with respect to the human-NeRF (Neural Body/HumanNeRF/UV-Volumes) and human-GS methods (Animatable Gaussians/D3GA/GoMAvatar).***
>
> Several works combine human faces/avatars and meshes, and those based on Gaussian representations are particularly relevant to this discussion. Examples include [f1-f5] - all of them can model dynamic avatars, most based on video data. However, these specialized works rely on human pose/face priors (e.g., FLAME fitting [f6]). We believe it would be valuable to explore the potential of adapting D-MiSo concepts in this context, building on existing works in the field. We envision Stage 1 focusing on effectively integrating such models in motion while the anchoring concept could be adapted to efficiently capture the small details of moving avatars. However, human avatars are a broad topic, with several notable and advanced works emerging each month. A thorough and accurate analysis of related works is required to fully investigate the best utilization and adaptation of our method in this context. We include the following papers in the Related Work and Future Work Sections.
>
> [f1] Li et al. 2024, “Animatable Gaussians: Learning Pose-dependent Gaussian Maps for High-fidelity Human Avatar Modeling”, in Proceedings of the IEEE/CVF CVPR
>
> [f2] Zielonka et al. 2023, “Drivable 3D Gaussian Avatars”, arXiv preprint arXiv:2311.08581.
>
> [f3] Wen et al. 2024, “Gomavatar: Efficient Animatable Human Modeling from Monocular Video Using Gaussians-on-Mesh,” in Proceedings of the IEEE/CVF CVPR
>
> [f4] Qian et al. 2024, "GaussianAvatars: Photorealistic head avatars with rigged 3d Gaussians," in Proceedings of the IEEE/CVF CVPR
>
> [f5] Xiang et al. 2023, "Flashavatar: High-fidelity digital avatar rendering at 300fps." arXiv preprint arXiv:2312.02214
>
> [f6] Li et al. 2017, “Learning a model of facial shape and expression from 4D scans”, in ACM Transactions on Graphics (Proc. SIGGRAPH Asia), 36(6)

---

> > ### Comment · Reviewer_qmWF · 2024-08-11
> >
> > Thanks for the rebuttal. It addressed my main concerns. However, I don't think the human-NeRF-based methods (such as Neural Body, HumanNeRF, and UV-Volumes) should be ignored. I believe it is fundamental for this paper to select some of these works and discuss the line of progress.

---

> > > ### Author Response · Authors · 2024-08-12
> > > **Authors’ response**
> > >
> > > We thank the Reviewer for the feedback. To provide a more balanced discussion, aside from already included Gaussian Splatting-based 3D avatars, we will include in our manuscript all the suggested NeRF-based works recognized for their following contributions:
> > >
> > > In [h1], authors introduce a novel human body representation, where learned neural representations at different frames share the same set of latent codes anchored to a deformable mesh. This approach allows for the natural integration of observations across frames and provides geometric guidance, leading to more efficient learning of 3D representations. Neural Body effectively addresses the challenge of ill-posed representation learning in scenarios with highly sparse monocular video views. Whereas HumanNeRF [h2] optimizes a volumetric representation of a person in a canonical T-pose, with a motion field that maps the canonical representation to each frame of the video via backward warps. The motion field is decomposed into skeletal rigid and non-rigid motions, handled by deep networks. HumanNeRF demonstrates significant performance improvements over previous methods and produces compelling free-viewpoint renderings from monocular video, even in challenging, uncontrolled capture scenarios. In addition, to address the high computational costs of NeRF rendering, [h3] proposes the UV-Volumes approach, which enables real-time, editable free-view video of human performers. By separating high-frequency appearance details from the 3D volume and encoding them into 2D neural texture stacks (NTS), the UV-Volumes allow for the use of smaller and shallower neural networks to achieve efficient 3D density and texture coordinate estimations while maintaining detailed 2D appearance capture.
> > >
> > > If this clarification adequately addresses the concerns raised, we kindly ask the Reviewer to consider raising their scores accordingly.
> > >
> > > [h1] Peng et al. 2021, "Neural Body: Implicit Neural Representations with Structured Latent Codes for Novel View Synthesis of Dynamic Humans," Proceedings of the IEEE/CVF CVPR
> > >
> > > [h2] Weng et al. 2022, "HumanNeRF: Free-Viewpoint Rendering of Moving People From Monocular Video," Proceedings of the IEEE/CVF CVPR
> > >
> > > [h3] UV-Volumes - Chen et al. 2023, "UV Volumes for Real-Time Rendering of Editable Free-View Human Performance," Proceedings of the IEEE/CVF CVPR

---

> > > > ### Comment · Reviewer_qmWF · 2024-08-12
> > > >
> > > > Thanks for the reply. It addressed my concerns. The progress from human-NeRF to human-GS should be discussed together, and at the same time, it should be more concluded and concise. Based on other reviews and the rebuttal, especially the training time comparisons and discussion against gs2mesh-based, cage-based, human-NeRF and human-GS methods, I raise my rating to Weak Accept.

---

> > > > > ### Author Response · Authors · 2024-08-12
> > > > > **Authors’ response**
> > > > >
> > > > > Thank you for dedicating your time to reviewing our manuscript. We believe that the revisions have given the work greater context, enhancing its value for the reader. We sincerely appreciate the decision to increase our score.

---

### Official Review · Reviewer_hxyh · 2024-06-26

**Soundness:** 3
**Presentation:** 3
**Contribution:** 3
**Rating:** 7
**Confidence:** 5

**Summary:**

This paper proposes a novel framework for modeling and editing dynamic scenes. The authors used a two-pass Multi-Gaussian approach to represent the entire scene. First, they obtained relatively stable Core Gaussians through initialization, and then used the Core Gaussians to drive Sub-Gaussians to fit the entire scene. To better edit motion, the authors parameterized each Gaussian with triangle soup. Experimental results show that this method not only matches the rendering quality of SC-GS but also enables the editing of more extreme large motions.

**Strengths:**

This paper is clear and easy to follow. The dynamic reconstruction method based on Gaussian splatting inherently has advantages in motion editing, but this direction has not been well-explored by the community. Therefore, I am very grateful to the authors for focusing on improving the motion editing capabilities of dynamic scenes and achieving impressive editing results (as shown in Fig. 4). The triangle-soup-based motion editing proposed by the authors really makes sense. Additionally, the honest comparison of rendering metrics in both synthetic and real scenes is also appreciated.

**Weaknesses:**

1. I think the following papers should also be cited, because they have made significant contributions to the early dynamic scenes reconstruction based on Gaussian splatting.:
   - (CVPR 2024) Spacetime Gaussian Feature Splatting for Real-Time Dynamic View Synthesis by Zhan Li et al.
   - (ICLR 2024) Real-time Photorealistic Dynamic Scene Representation and Rendering with 4D Gaussian Splatting by Zeyu Yang et al.
2. L136, The concept of anchor Gaussian originally comes from Scaffold-GS [1], not from Spec-Gaussian. It would be even better if Scaffold-GS could be cited.
3. It would be even better if Deformable-GS [2] were included in the comparison of quantitative metrics (such as Tabs. 1-2), as it is the first deformation-based dynamic Gaussian splatting method and serves as the baseline for SC-GS.
4. I think the ablation study is not thorough enough. Although the ablation on batch size is appreciated, I am unclear about the roles of other components of the method. Corresponding ablation studies are needed to make the paper more robust. For example, I am very interested in understanding the impact of the `Sub-Rot Network` mentioned in L218 on the rendering metrics.



[1] Tao Lu, Mulin Yu, Linning Xu, Yuanbo Xiangli, Limin Wang, Dahua Lin, and Bo Dai. 2023. Scaffold-GS: Structured 3D Gaussians for View-Adaptive Rendering. arXiv preprint arXiv:2312.00109 (2023)

[2] Ziyi Yang, Xinyu Gao, Wen Zhou, Shaohui Jiao, Yuqing Zhang, and Xiaogang Jin. Deformable 3d gaussians for high-fidelity monocular dynamic scene reconstruction. arXiv preprint arXiv:2309.13101,2023.

**Questions:**

A small tip: In Tab. 1, the authors used the metrics from the SC-GS paper. However, SC-GS did not use a consistent background; for example, the `Bounce` (and maybe `Trex`) scene used a white background. The authors clearly used a black background consistently. Although the reported metrics in the table show a slight decrease compared to SC-GS, I greatly appreciate the authors' honesty.

**Limitations:**

Please refer to the weakness part.

---

> ### Author Rebuttal · Authors · 2024-08-05
>
> We thank the Reviewer for the vote of confidence in our manuscript and for the in-depth feedback and suggestions, which we are happy to incorporate and feel will improve the manuscript. We respond to all the questions and concerns raised in the answers below.
>
> ***W1: I think the following papers should also be cited, because they have made significant We contributions to the early dynamic scenes reconstruction based on Gaussian splatting …***
>
> Thank you for suggesting this; we will cite these papers in our manuscript. Method [b1] proposes approximating the spatiotemporal 4D volume of a dynamic scene by optimizing a collection of 4D primitives with explicit geometry and appearance modeling. This method uses 4D Gaussians parameterized by anisotropic ellipses and view-dependent, time-evolved appearances represented by 4D spherical harmonics coefficients. In [b2], a novel scene representation called Spacetime Gaussian Feature Splatting has been introduced. This approach extends 3D Gaussians with temporal opacity and parametric motion/rotation, enabling the capture of static, dynamic, and transient scene content. Additionally, the method incorporates splatted feature rendering to model view- and time-dependent appearances while maintaining a compact representation size. The method is notable for its high rendering quality and speed while also being storage-efficient. We agree that both these works have significantly contributed to the reconstruction of dynamic scenes and align well with the topic under discussion. **Both of them will be included in our manuscript.**
>
> [b1] Li et al., 2024, “Spacetime Gaussian Feature Splatting for Real-Time Dynamic View Synthesis,” Proceedings of the IEEE/CVF Conference on Computer Vision and Pattern Recognition
>
> [b2] Yang et al., 2024. “Real-time Photorealistic Dynamic Scene Representation and Rendering with 4D Gaussian Splatting”, Proceedings of the International Conference on Representation Learning
>
> ***W2: L136, The concept of anchor Gaussian originally comes from Scaffold-GS [1], not from Spec-Gaussian. It would be even better if Scaffold-GS could be cited.***
>
> Thank you for this excellent comment. In [c1], the authors introduce the concept of anchor points to tackle the problem of overfitted models caused by redundant Gaussians. Scaffold-GS addresses this issue by distributing local 3D Gaussians according to anchor points and predicting attributes based on viewing direction and distance. This approach reduces redundancy, enhances scene coverage, and maintains high-quality rendering with improved robustness to view changes. **Therefore, we will provide an extended description of [c1] in Related Works.**
>
> [c1] Lu et al., “Scaffold-GS: Structured 3D Gaussians for view-adaptive rendering,” Proceedings of the IEEE/CVF, CVPR 2024
>
> ***W3: It would be even better if Deformable-GS [2] were included in the comparison of quantitative metrics (such as Tabs. 1-2), as it is the first deformation-based dynamic Gaussian splatting method and serves as the baseline for SC-GS.***
>
> The model Deformable-GS [d1] is indeed a prototype of the SC-GS model. Deformable-GS is one of the first works targeting deformable/dynamic scenes. It introduces an MLP-based deformation network to adapt Gaussian parameters to a given timestep. Below are the PSNR results of our comparison with SC-GS and Deformable-GS, which we will include in our article.
>
> |Dataset:|Hook|Jumpin|Trex|Bounce|Hell|Mutant|Standup|
> |---|---|---|---|---|---|---|---|
> |Deformable-GS|37.42|37.72|38.10|41.01|41.54|42.63|44.62|
> |SC-GS |39.87|41.13|41.24|44.91|42.93 |45.19 |47.89|
> |D-MiSo (our) |38.13|42.05|40.88 |41.49 |41.49 |44.38|47.66|
>
>
> [d1] Yang et al., 2023, “Deformable 3D Gaussians for High-Fidelity Monocular Dynamic Scene Reconstruction”, arXiv preprint arXiv:2309.13101.
>
> ***W4 I think the ablation study is not thorough enough. Although the ablation on batch size is appreciated, I am unclear about the roles of other components of the method. Corresponding ablation studies are needed to make the paper more robust. For example, I am very interested in understanding the impact of the Sub-Rot Network mentioned in L218 on the rendering metrics.***
>
> The Sub-Rot Network, despite being a simple single-layer network, plays a crucial role in the 2nd stage. We appreciate it being highlighted for further analysis, as this allows us to demonstrate its significance. Below is a numerical comparison of PSNR for models with and without the Sub-Rot Network. Furthermore, we would like to emphasize that due to its shallow architecture, incorporating the Sub-Rot Network does not significantly impact training time. We decided to present the results on the *jumpingjack* dataset distinguishing between batch = {4, 8}. The experiments we performed using the RTX4090 GPU.
>
> |   | batch = 4  |batch = 4   |batch = 8   |  batch = 8 |
> |---|---|---|---|---|
> | Sub-Rot Network  | with  |without   | with  |  without |
> |PSNR | 40.42 | 39.98 | 41.65 | 41.27|
> |Training time [h] | 1:18|1:12|2:08|1:45|
> |Rendering time [fps] |175 |186|190|259|
>
> Since the Sub-Rot Network used is not deep and has only a single layer, it does not visibly affect the training time. Additionally, Reviewer 1 (u5NP) pointed us to consider the role of a number of Sub-Gaussian/Core-Gaussian. We agree that this is an interesting analysis, and we will incorporate it in the revised manuscripts.
>
> ***Q1: A small tip: In Tab. 1, the authors used the metrics from the SC-GS paper. However, SC-GS did not use a consistent background; for example, the Bounce (and maybe Trex) scene used a white background. The authors clearly used a black background consistently. Although the reported metrics in the table show a slight decrease compared to SC-GS, I greatly appreciate the authors' honesty.***
>
> Thank you for raising this point. We believe it is important to use a unified framework for evaluation. Therefore, following most papers, we introduced results obtained on a black background.

---

> > ### Comment · Reviewer_hxyh · 2024-08-08
> > **Keep positive**
> >
> > Thanks to the authors for their response. I am satisfied with the rebuttal and will maintain my positive evaluation. There are a few points to note for the release version:
> > - The resolution of Deformable-GS is 800x800, while SC-GS and D-MiSo are 400x400. I don't think it's necessary to align them completely, but it would be better to clarify this.
> > - For SC-GS FPS measurement, KNN should be fixed, and there is no need to query for every iteration.
> > - I suggest that the authors discuss the differences and connections between D-MiSo and GaMeS.

---

> > > ### Author Response · Authors · 2024-08-08
> > > **Authors’ response**
> > >
> > > We thank the Reviewer for the valuable comments and are keen to follow up on the provided suggestions:
> > >
> > > - We will add information on image resolution and include a table with 800x800 resolution in the Appendix.
> > > - We agree that KNN can be fixed and will include a comment to that effect in the paper.
> > >
> > > - We will discuss the differences and connections between D-MiSo and GaMeS in the main paper.

---

### Official Review · Reviewer_u5NP · 2024-06-29

**Soundness:** 3
**Presentation:** 2
**Contribution:** 2
**Rating:** 5
**Confidence:** 2

**Summary:**

This paper introduces a Dynamic Gaussian Splatting representation that allows for easier object shape editing at test time. This is achieved through the use of Dynamic Multi-Gaussian Soup (D-MiSo), a mesh-inspired multi-gaussian system. Specifically, a GS is divided into two components: Core-Gaussian, which models the transformation of groups of sub-gaussians, and sub-Gaussian, which handles the final geometry and color. This enables different levels of editing by either changing the core-Gaussian or sub-Gaussians for more fine-grained control.

**Strengths:**

1) The paper demonstrates substantial improvement in the level of control and editability of Gaussian Splatting (GS) at test time, while maintaining competitive PSNR/SSIM scores.
2) I appreciate the inclusion of mesh visualization estimates (Figures 9 and 11), which make the method more convincing.
3) The method is tested on various datasets with varying complexity and shows good performance.

**Weaknesses:**

I find the paper's writing style difficult to follow and some sentences do not even make sense. Additionally, there are numerous components involved in the pipeline that are challenging to keep track of: multi-Gaussian, core-Gaussian, sub-Gaussian, core-triangle soup, multi-triangle soup, and sub-Gaussian soup (which may be the same thing?). I would appreciate a revision in writing to improve clarity.

**Questions:**

1) Is there a strategy for selecting hyperparameters such as the number of Gaussian and Sub-Gaussian components?
2) It is unclear why the proposed method would perform worse than SC-GS on metrics like PSNR/SSIM for NeRF-DS and D-NeRF. Does this suggest that increased editability comes at the cost of model performance, or is it simply a matter of hyperparameter tuning?

**Limitations:**

The authors have discussed the limitations of their method, but I was unable to locate a statement regarding the broader impacts of the proposed method in the Limitations section, as claimed by the author.

---

> ### Author Rebuttal · Authors · 2024-08-05
>
> We thank the Reviewer for the constructive remarks regarding our manuscript, to which we responded next to the posted questions below. We will revise the manuscript in accordance with the raised concerns.
>
> ***W1 “I find the paper's writing style difficult to follow …  I would appreciate a revision in writing to improve clarity.”***
>
> Thank you for this comment. We appreciate the complex nature of our manuscript, and we will edit all the relevant parts for clarity. In the paper, we introduced a Multi-Gaussian component, which is comprised of a Core-Gaussian "parent" and Sub-Gaussian "child," each parameterized by a triangle. Accordingly, the Core-Triangle corresponds to the Core-Gaussian, and the Sub-Triangle corresponds to the Sub-Gaussian. The terms "Triangle" or "Gaussian" are used contextually:
>
> - When discussing rendering or properties such as color and transparency, we use "Gaussian."
>
> -  When referring to parameterized Gaussians and their modifications (e.g., location, scale, rotation), we use "Triangle."
>
> Fig. 3 illustrates these differences. **We will explain all these objects and clarify them in the Introduction.**
>
> ***Q1:  Is there a strategy for selecting hyperparameters such as the number of Gaussian and Sub-Gaussian components?***
>
> Our model operates on several hyperparameters, primarily derived from the basic Gaussian Splatting (GS) framework. Given the introduction of additional stages and the multi-Gaussian component, one of the new hyperparameters is the number of Core-Gaussians and Sub-Gaussians. We are pleased to see this novel aspect being acknowledged. The number of Core-Gaussians is determined automatically in stage 1, utilizing the pruning mechanism implemented in GS.
> The Table below illustrates the training time corresponding to these parameters using the *jumpingjacks* dataset as an example. During the experiments, we used the RTX4090 GPU and densification until 5000 iterations (1st stage). We can see the speed of the 1st stage, which is used for Core-Gaussians preparation. Moreover, too few Core-Gaussians can cause a drop in quality (PSNR metric). The sub-rot network is shallow enough (it is a single layer in all of our experiments) that the increased cost of training time is not apparent. Therefore, the experiments suggest the number of Core-Gaussians is sufficient after the first phase. In all cases, a larger number of Sub-Gaussians improved the quality of renders.
>
> |n_sub|1|10|25|1|10|25|1|10|25|
> |---|---|---|---|---|---|---|---|---|---|
> |Iteration - start of deform network|2000|2000|2000|3000|3000|3000|5000|5000|5000|
> |N_core  after 1st stage|3290|3340|3434|2841|2974|2936|1803|1798|1807|
> |PSNR|38.94|41.35|41.65|38.52|41.14|41.41|33.63|37.14|37.47|
> |Train time 1st stage [h]|00:05:14|00:04:48|00:05:17|00:04:17|00:04:33|00:04:16|00:03:27|00:03:02|00:03:21|
> |Train time 2nd stage [h]|1:51:17|1:44:21|2:03:25|1:39:39|1:53:18|1:50:25|1:48:54|1:41:12|1:56:28|
> |Sum: Train time [h]|1:56:31|1:49:09|2:08:42|1:43:56|1:57:51|1:54:41|1:52:21|1:44:14|1:59:49|
>
> ***Q2: It is unclear why the proposed method would perform worse than SC-GS on metrics like PSNR/SSIM for NeRF-DS and D-NeRF. Does this suggest that increased editability comes at the cost of model performance, or is it simply a matter of hyperparameter tuning?***
>
> Our main contribution is the ability to edit a dynamic scene better and more efficiently. As pointed out, there is generally a trade-off between editing and the quality of the reconstruction. However, **in our model, such effects are minimal**. Consequently, we obtained markedly worse results on synthetic datasets (Tab 1 in our draft paper) **and better results achieved on full scenes** (Tab. 3 and Tab. 4  in our draft paper). We tentatively believe this is caused by the suboptimal algorithm used to choose core points in SC-GS. SC-GS uses a heuristic approach based on distances to select core points, which might be problematic on a large scale. Our solution uses classical GS optimization procedures to establish the Gaussian number and size. Therefore, D-Miso covers 3D scenes better.
>
> ***L1: The authors have discussed the limitations of their method, but I was unable to locate a statement regarding the broader impacts of the proposed method in the Limitations section, as claimed by the author.***
>
> Our model significantly improves rendering quality and advances 3D scene reconstruction and rendering, impacting multiple domains by enabling more realistic and efficient 3D modeling and animation. This technology could enhance VR/AR experiences [a1], robotics [a2], and medical imaging [a3]. It could also be used for interactive education [a3], scientific visualization, and a plethora of other commercial applications like product design and real estate [a4]. We will edit our manuscript to include these considerations and potential application fields of our method.
>
> [a1] Linning et al. 2023, “VR-NeRF: High-Fidelity Virtualized Walkable Spaces”, SIGGRAPH Asia 2023 Conference Papers
>
> [a2] Lisus et al., 2023, "Towards Open World NeRF-Based SLAM," 2023 20th Conference on Robots and Vision (CRV), Montreal, Canada, pp. 37-44, doi: 10.1109/CRV60082.2023.00013
>
> [a3] Huang et al. 2013, “Exploring Learner Acceptance of the Use of Virtual Reality in Medical Education: A Case Study of Desktop and Projection-Based Display Systems,” Interactive Learning Environments 24 (1): 3–19.
>
> [a4] Li et al. 2022, “Deep Learning of Cross-Modal Tasks for Conceptual Design of Engineered Products: A Review”, ASME 2022 International Design Engineering Technical Conferences and Computers and Information in Engineering Conference

---

> > ### Comment · Reviewer_u5NP · 2024-08-11
> >
> > Thank you for your comprehensive answer. I am still curious why your method peforms worse on synthetic data which should be easier?

---

> > > ### Author Response · Authors · 2024-08-12
> > > **Authors’ response**
> > >
> > > It is a non-trivial question requiring substantial consideration; thank you for posting it.
> > >
> > >  - **D-MiSo vs SC-GS performance on synthetic data:**
> > >
> > > The synthetic dataset from the paper describing D-NeRF [11] is the first and most explored dataset of dynamic 3D objects. **Modern methods on the D-NeRF dataset produce similar results with small differences caused mainly by parameter optimization.** Results on the D-NeRF dataset are proof that our model has attained the presently possible optimal level of rendering quality (Reviewer 2: “Experimental results show that this method not only matches the rendering quality of SC-GS but also enables the editing of more extreme large motions.”). **Consequently, our model's main contribution is better editing quality.**
> > >
> > > - **D-MiSo vs SC-GS performance on real data:**
> > >
> > > In the SC-GS paper, the authors discuss the use of control points. Each control point influences the closest Gaussian components. The number of control points is a crucial hyperparameter. As detailed in Section S1.1 of the SC-GS supplementary material, determining the **optimal** number of control points is essential for achieving accurate reconstructions. However, following the SC-GS supplementary material, we think that in real scenes with complex backgrounds, identifying the appropriate number of control points may be challenging. Increasing the number of control points does not necessarily lead to better performance due to optimization challenges.
> > >
> > > In contrast, our D-MiSo model automatically covers space, and no additional hyperparameters are introduced. The number of Core Gaussians is automatically found by the classical Gaussian Splatting optimization procedure in the first part of optimization. Later, each scene part can be modified independently.  As a result, our model not only achieves high-quality reconstructions but also facilitates more effective editing, thus balancing reconstruction quality with editing capability. In essence, our solution does not use additional parameters, and our model builds a natural division of objects by the Core Gaussian component. It also allows for more straightforward modification. **Consequently, our model's main contribution is better editing quality.**
> > >
> > > [11] Pumarola, Albert, et al. "D-NeRF: Neural radiance fields for dynamic scenes." Proceedings of the IEEE/CVF Conference on Computer Vision and Pattern Recognition. 2021.

---

> > > > ### Comment · Reviewer_u5NP · 2024-08-12
> > > >
> > > > Thank you for your answer. All my concerns are addressed.

---

> > > > > ### Author Response · Authors · 2024-08-12
> > > > > **Authors’ response**
> > > > >
> > > > > We appreciate the careful and constructive evaluation of our work and engaging in the discussion.

---

### Author Rebuttal · Authors · 2024-08-05

We thank the Reviewers for their excellent comments and constructive remarks regarding our paper, as well as for their positive feedback. We are also thankful for noticing the main contribution of our model, which is “enabling the editing of more extreme large motions” that “matches the rendering quality of SC-GS”, as underlined by Reviewer 2. The main two strands of the received feedback concern the need for further ablation studies and providing a more comprehensive literature review. We feel that these changes will enhance our paper, and we will incorporate them into our manuscript.

**Ablation studies and hyperparameter analysis**

Most of the comments received concerned the need for an ablation study. First, we contrasted our model with and without the Sub-Rotation Network, as shown in the Table below, utilizing the commonly used *jumpingjacks* dataset. **These results show that Sub-Rot Network is crucial to obtaining SOTA PSNR. With Sub-Rot Network, we obtain approximately 0.5 PSNR crucial to obtain the reconstruction of small elements** (like human fingers in the *jumpingjacks* dataset).


|   | batch = 4  |batch = 4   |batch = 8   |  batch = 8 |
|---|---|---|---|---|
| Sub-Rot Network  | with  |without   | with  |  without |
|PSNR | 40.42 | 39.98 | 41.65 | 41.27|
|Training time [h] | 1:18|1:12|2:08|1:45|
|Rendering time [fps] |175 |186|190|259|

We also present how the size of the network influences training time and render speed in the next Table utilizing the commonly used *jumpingjacks* dataset. **These results show that deformable networks and Sub-Rot Networks are not costly in terms of rendering time, resulting in real-time rendering.**
In practice, batch size has a higher impact than deformation network depth.

|Deformation network depth (numbers of layers)|4|6|8|10|
|---|---|---|---|---|
Batch:|4|4|4|4|
|PSNR|40.91|40.75|40.42|40.26|
|Training time [h]|1:21|1:22|1:18|1:31|
|Render time [fps]|138|167|175|144|
Batch:|8|8|8|8|
PSNR|41.86|42.01|41.65|41.37|
Training time[h]|2:00|1:58|2:08|2:07|
Render speed(fps)|227|221|190|192|

The following Table presents the training time and storage cost, as well as the FPS for rendering for each dataset. **These results suggest that our model is memory efficient and that the time required to train it is minimal.**  We performed the experiments for batches 4 and 8 to show the effect of batch size. In all cases, storage costs decreased, and performance improved with a trade-off of increased training time.


Dataset:| Hook|Jumpin|Trex|Bounce|Hell|Mutant|Standup|
|---|---|---|---|---|---|---|---|
Batch:|4|4|4|4|4|4|4|
PSNR|37.77|40.42|39.56|40.63|41.44|43.38|46.07
Time 1st stage|00:02:48|00:02:22|00:02:41|00:03:12|00:02:27|00:03:46|00:03:17
Time 2nd stage |1:40:15|1:16:18|2:14:09|1:43:08|1:01:27|1:48:49|1:19:32
Storage cost|76MB|53.5MB|122MB|131MB|27MB|73MB|43MB
Rendering time  (fps)|138|175|90| 123| 185 | 138 | 169
Batch:|8|8|8|8|8|8|8
|PSNR|38.07|41.65|40.74|40.55|41.59|44.40|47.22|
|Time 1st stage [h] |00:05:30|0:05:17|00:05:38|00:05:16|00:04:28|00:05:51|00:05:56|
|Time 2nd stage [h] |2:27:08|2:03:25|2:42:22|2:40:30|1:43:33|2:18:43|2:12:12|
|Storage cost|32MB|24MB|51MB|80MB|16MB|28MB|20MB|
|Rendering time(fps)|192|190|143|153|205|188|194

**Related Works**

Another repeated feedback concerned extending the body of referenced prior works. In line with this feedback, we will reorganize and extend the literature review to include the suggested related works and make it more complete. We are also keen to acknowledge that these additional papers present lower scores than SG-CS and D-Miso. Moreover, we reckon that our D-Miso model has a stronger ability to edit dynamic scenes.

---

### Decision · Program_Chairs · 2024-09-25

**Decision:**

Accept (poster)

**Comment:**

This paper introduces a Dynamic Gaussian Splatting representation that enables easier object shape editing at test time. There is an overall consensus among reviewers that this is an interesting submission and should be accepted (2x Borderline Accept, 1x Weak Accept, and 1x Accept). Therefore, I recommend acceptance.

For the camera-ready version, please polish the paper to make it more reader-friendly and include the new experiments and discussions.